# Comparing thaw probing, electrical resistivity tomography, and airborne LiDAR to quantify lateral and vertical thaw in rapidly degrading boreal permafrost

Thomas A. Douglas[1], Mark Torre Jorgenson[2], Taylor Sullivan[1], Caiyun Zhang[3]

[1]U.S. Army Cold Regions Research and Engineering Laboratory, Fort Wainwright, Alaska 99703, United States

[2]Alaska Ecoscience, Fairbanks, Alaska 99709, United States

[3]Department of Geosciences, Florida Atlantic University, Boca Raton, Florida 33431, United States

*Correspondence to*: Thomas A. Douglas (Thomas.a.douglas@usace.army.mil)

**Abstract.** Permafrost thaw across Earth's high latitudes is leading to dramatic changes in vegetation and hydrology. We undertook a two-decade study near Fairbanks, Alaska to measure permafrost thaw and associated ground surface subsidence via field-based and remote-sensing techniques. Our study focused on transects representing an unburned area and three fire scars (1988, 2001, and 2010). Three types of permafrost quantification were used. First, repeat measurements of ground-surface elevation and depth to the top of near-surface permafrost were made over an 8-to-21-year period at different sites. Widespread near-surface permafrost degradation occurred between 2004 and 2020 with top-down thaw of near surface permafrost doubling from 18% to 36%. Permafrost aggradation was almost completely absent by 2020. Second, we calculated rates of top-down versus lateral thaw using airborne LiDAR from 2014 and 2020. Lateral thaw of tabular permafrost bodies and development of unfrozen zones between the bottom of the seasonally frozen layer and the top of near-surface permafrost (taliks) were evident. Third, electrical resistivity tomography (ERT) measurements from 2012 and 2020 supported surface-based thaw observations and allowed subsurface permafrost mapping up to 20 m deep. No single method provides all the information needed to adequately assess permafrost change. For example, frost probing yields insight into top-down thaw, LiDAR allows identification of vertical and lateral subsidence, and ERT identifies the presence/absence of permafrost at 10s of meters depth. Future applications of these methods should focus on relating surface and subsurface variables measured on the ground with information that can be remotely sensed across broad regions.

## 1. Introduction

Permafrost is warming and degrading across Earth's high latitudes. In interior Alaska this includes top-down thaw of near-surface permafrost (Douglas et al., 2021), increased permafrost temperatures and the formation of unthawed zones (taliks; Farquharson et al., 2022), and lateral expansion of thawed areas (Jorgenson et al., 2020). Mean annual air temperatures in the region have increased ~3C since the 1970s (Douglas et al., 2025) and summers are getting wetter (Jorgenson et al., 2020). The

ongoing permafrost degradation affects hydrology (Marshall et al., 2021), ecological processes (Foster et al., 2019; Mekonnen et al., 2019), the carbon cycle (Douglas et al., 2014) and infrastructure (Hjort et al., 2022). With warming projected to accelerate over coming decades the spatial extent of permafrost thaw is expected to increase (Wolken et al., 2011).

In warm permafrost regions like interior Alaska surface vegetation and organic matter insulate frozen ground from summer warmth (Shur and Jorgenson, 2007). Removal of this protective cover by wildfire, infrastructure development, and/or climate-driven processes affects the ground thermal regime and leads to accelerated permafrost thaw (Brown et al., 2015; Johnstone et al., 2020; Jorgenson et al., 2022). The relative contributions of press (climate warming) and pulse (fire or infrastructure development) disturbances on permafrost thaw are difficult to quantify over space and time, especially when they are compounded.

Efforts have been made to pinpoint hotspots of permafrost thaw and calculate rates of ground surface subsidence using repeat airborne LiDAR and other imagery products (Jones et al., 2013; Jorgenson et al., 2022; Zhang et al., 2023). These studies rely on differences in surface elevation or changes in vegetation cover over time to identify where permafrost extent has likely changed. Linking repeat geophysical measurements with surface and subsurface surveys is valuable for mapping three dimensional changes in permafrost extent at mountain (Mewes et al., 2017; Buckel et al., 2022) and lowland sites (Lewkowicz et al., 2011; Douglas et al., 2016; Minsley et al., 2023). Combining ground-based geophysical measurements with airborne remote sensing observations provides an effective means for quantifying three-dimensional permafrost thaw (Minsley et al., 2015; Uhlemann et al., 2021). These typically include site-based surface measurements of the depth of maximum summer season thaw (active layer) and cores of permafrost to quantify ice content.

Broad spatial application of terrestrial geophysical measurements with ground truthing is time intensive. As such, great uncertainty remains in our ability to quantify local rates of permafrost thaw linked to ground surface changes that can be remotely sensed over large areas or over time. Increasingly, machine learning geospatial analyses have shown utility in projecting established connections between vegetation and seasonal thaw from the plot (~1 km$^2$; Zhang et al., 2024) to the regional (~100 km$^2$; Zhang et al., 2021; 2023; Brodylo et al., 2024) scales. However, these analyses have not been used with high resolution ground surveys or geophysical measurements to measure subsurface permafrost degradation over time.

The Tanana Flats in central Alaska, where we conducted our study, has been subject to extraordinarily rapid permafrost degradation in response to climate warming, groundwater movement, and fire disturbance. The climate has been warming since the Little Ice Age but has been accelerating since the 1980s causing widespread permafrost degradation (Jorgenson et al. 2022). Extreme precipitation events have been another important climatological driver leading to permafrost thaw (Douglas et al. 2020). Near-surface groundwater movement through fens and subsurface gravels have created an extensive network of floating-mat fens (Racine and Walters 1994) and the heat associated with this groundwater contributes to permafrost degradation (Jorgenson et al. 2020). Fires have burned much of the Tanana Flats and led to immediate thaw and surface collapse in some areas (Nossov et al. 2015, Douglas 2015, Jorgenson et al. 2025). The extensive and rapid permafrost degradation is causing a rapid ecological transformation from lowland black spruce and birch forests to thermokarst bogs and fens (Racine et al. 1998, Jorgenson et al. 2001, Lara et al. 2015, Jorgenson et al. 2022, Jorgenson et al. 2025). The repeated

formation and degradation of ecosystem-driven permafrost in boreal Alaska in response to the interacting drivers, however,
greatly complicates the assessment of permafrost dynamics (Jorgenson et al. 2022). In this study we focus on how best to
quantify and monitor permafrost degradation rather than assess the drivers of change and ecological changes.

The focus of this work was to measure three dimensional rates of change in permafrost extent in a lowland area containing
ice-rich permafrost in Interior Alaska. Objectives were to: (1) quantify permafrost degradation from 2012 to 2020 across a
range of fire scars (1988, 2001, 2010) and an unburned site using repeat ground-based surveys; (2) use repeat airborne LiDAR
to map thaw stage and calculate the extent, volume, and rates of permafrost loss from vertical versus lateral degradation; and
(3) use repeat electrical resistivity tomography (ERT) to measure changes in subsurface permafrost conditions. We focus on
thaw probing, ERT, and airborne LiDAR methods because they are among the widest applied techniques to measure permafrost
degradation. This study design allowed us to evaluate the reliability of the three different approaches for quantifying thaw
stages.

## 2 Study sites and methods

### 2.1 Study site climatology, permafrost geomorphology, and fire history

Our study focused on four transects located along the northern edge of Tanana Flats, a lowland underlain by discontinuous
permafrost that stretches from 5 kilometers south of Fairbanks, Alaska to the north slopes of the Alaska Range ~70 km further
south (Fig. 1). Tanana Flats is a broad valley covering approximately 6,000 km$^2$ that spreads northward from the Alaska Range
to the Tanana River. Geologically, the Tanana Flats is a complex of fluvial deposits associated with a large outwash fan in the
western portion of the area and braided floodplain deposits in the northern and eastern portion of the area (Jorgenson et al.,
1999; Walters et al., 1998). Much of the area has a typical stratigraphic sequence of peat (0.5-1.5 m), aeolian silt (2-3 m), and
alluvial sand and gravel (Jorgenson et al., 1999; Brown et al. 2015). Permafrost covers about 44% of the area (Jorgenson et
al., 2001). Permafrost is mainly epigenetic and formed during downward freezing. Excess ice content in permafrost below
birch forests can be greater than 50% while ice contents in the black spruce stands are typically closer to 20% (Brown et al.,
2015; Jorgenson et al., 2001; Jorgenson et al. 2025; Walters et al., 1998). Deep boreholes found permafrost at depths ranging
from 7.3 m (Ferrick et al., 2008) to 47 m (Jorgenson et al., 2001), while ERT indicated minimum thicknesses of >20 m were
common (Douglas et al., 2015). The mean annual temperatures near the top of permafrost for undegraded permafrost are
between 0 and -1 °C (Brown et al. 2015; Jorgenson et al. 2025). Active-layer thickness above permafrost typically ranges from
50 to 75 cm in birch and spruce forests (Brown et al., 2015).

The area is characterized by a heterogeneous patchwork of forested areas underlain by discontinuous permafrost up to 50
m thick (Chacho et al., 1995). Numerous bogs and fens dot the landscape. They are not generally underlain by permafrost but
their outside margins are delineated by frozen ground (Douglas et al., 2016). From the air these wetland features are readily
apparent, however, the morphology of their subsurface permafrost boundaries, particularly along their margins, is not well
known. More detailed descriptions of the ecology, hydrogeology, and permafrost thermal state of our sites are provided
elsewhere (Brown et al., 2015; Douglas et al., 2016; Lara et al., 2016; Jorgenson et al., 2022).

The regional climate is continental with a mean annual temperature of -2.4 °C and mean monthly temperatures ranging from 16 °C in summer to -22 °C in winter. Annual extremes range from -51 °C to 29 °C (Jorgenson et al., 2020). Typical mean annual precipitation is 28 cm water equivalent with 45% of this as snow (Liston and Hiemstra, 2011). Based on decadal mean annual temperatures at the Fairbanks International Airport the area warmed ~2.3 °C between the 1930s-1940s and 2010-2020. Over that same timeframe mean summer temperatures (May 1 to October 10) warmed ~1.7 °C while mean winter temperatures (October 11-April 30) warmed ~3 °C (Douglas et al., 2024).

More detailed information on the regional climate, with a focus on Tanana Flats, is provided in Supplemental Information Figs. 1 and 2. The period between 2012 and 2020 (when the majority of the field measurements were collected, including repeat ERT) included three of the ten warmest/wettest summers and three of the ten warmest wettest winters in the ~100-year record. However, no top ten summers or winters for cool and dry or cold and low snow, favorable to permafrost stability or aggradation, occurred during the same timeframe. Two of the five wettest summers in the entire meteorological record (2014 and 2016) and two of the top three highest mean annual air temperatures (MAATs) were also recorded during the study period. This provided conditions favorable to permafrost thaw.

The dominant forest cover on Tanana Flats includes deciduous Alaska paper birch (*Betula neoalaskana*) and aspen (*Populus tremuloides*) mixed with pure white spruce (Picea glauca) or mixed white and black spruce (*Picea Mariana)*. Ground cover is dominated by *Sphagnum* spp. In poorly drained areas feather mosses (*Pleurozium schreberi, Hylocomnium splendens*) are common. This land cover is well suited to protect permafrost from warm summers, however, it is also subjected to fire return intervals of 50-130 years with more frequent fires in black spruce stands (Johnstone et al., 2010; Brown and Johnstone, 2012; Douglas et al., 2014; Brown et al., 2015; Potter and Hugny, 2020).

The Tanana Flats lowland has experienced numerous wildfires and we established transects to represent high severity fires in the summers of 1988 (TF88, 64.734 °N, 147.826 °W), 2001 (TF01, 64.644 °N, 148.295 °W), and 2010 (TF10, 64.716 °N, 148.010 °W). Transects were initially studied to assess the effects of fire on permafrost (Nossov et al., 2013): TF88 (200 m; first surveyed in 2012) is in an area burned in ~1950 that reburned in 1988; TF01 (established as a 320 m transect in 2012) is in an area burned in 2001, and TF10 (initially 100 m in 2011, extended to 200 m in 2012) is in an area burned in 2010. Transects were positioned to cross a range of permafrost and non-permafrost ecotypes and TF01 and TF10 were extended to more adequately represent different cover types. We also included an unburned site (T1, 64.722 °N, 147.959 °W) that has not burned in recent years (~1950s-present) for comparison. Supplemental Information Figs. 3 and 4 provide photographs of the field sites. T1 (initially 100 m in 1995, extended to 200 m in 2012) was established during ecological land surveys (Jorgenson et al. 1999). In Douglas et al. (2016) T1 is referred to as "1930". Two of the sites (T1 and TF88) were relatively ice-rich with thick peat and silts extending down 34 m whereas the other two study sites (TF01 and TF10) had sand and gravel at relatively shallow depths and lower ice contents.

## 2.2. Data collection, processing, and analysis

### 2.2.1 Field sampling design and measurements

Along our four transects we measured surface topography, seasonal thaw depth, areal extent and depth of standing water, and ERT in the fall of 2012. Sites were revisited in late summer almost yearly until 2020 to repeat measurements of ground-surface elevation, thaw depths, and water depth, and to download temperature loggers. ERT lines and elevation surveys were remeasured in 2020.

Topographic surveys of ground- and water-surface elevations usually were made using an auto-level and stadia rod at 1-m intervals along the four permanently marked transects. We estimate measurement accuracy to be within 3 cm for firm ground and within 10 cm for soft floating mats in bogs and fens. A survey-grade, differential global position system (DGPS) was used in 2012 at TF10 and in 2013 at TF01, TF10, and TF50 to determine ground elevations using a 15-second observation time at each 1-m interval. The data were post-processed using data from a base station in Fairbanks. We estimate the accuracy to be mostly within 10 cm in open areas and 30 cm in forested areas. For auto-level surveying, elevations were calculated relative to permanent benchmarks, where the elevations were determined through repeated measurements.

Repeated thaw probing using a metal rod in late summer (September) quantified changes in the top of near-surface permafrost. Maximum probing depths varied from 2.5 to 4.0 m depending on the number of extensions used, occurrence of gravel, and stickiness of unfrozen silts. Accuracy of the field probing depends on soil materials, unfrozen-water content in partially thawed soils, and depth. As the soils within the top 3 m were mostly peat and silt, shallow (<1 m) probing typically hit a hard refusal boundary (indicating frozen conditions) for stable permafrost; for these conditions we consider the accuracy to be within 3 cm (given a "soft" mossy and litter-rich ground surface). Occasionally, in partially degrading permafrost near the surface (within 1.5 m) the probe encountered frozen ground with substantial unfrozen water content with refusal gradually becoming harder across a ~20 cm transition zone; we consider the accuracy under these conditions to be within 20 cm. Deep (2 to 5 m) probing is more problematic as friction along the entire probe increases: for these conditions we believe we can reliably detect a frozen boundary within 2 m, but by 5 m the friction/stickiness is such that probing becomes unreliable. At these depths, we are confident in the determination of unfrozen conditions, but have only low to moderate confidence assigning a refusal to be the result of frozen conditions. For these situations we repeatedly raise and thrust the probe downward and used our best judgement to assign frozen/unfrozen status.

Differentiating persistent seasonal frost from very thin permafrost is a challenge, particularly when only frost probe measurements are available (i.e. no boreholes). Here we are confident that frost <30 cm thick at the end of summer is seasonal frost and frost >50 cm thick is permafrost; for in between thicknesses we have low confidence as to whether it is seasonal or permanent frost. In some areas we noted a thin frost layer (30 to 50 cm) to persist for 2 to 3 seasons, and we refer to this as multi-year frost. This multi-year frost is a common problem for probing ecosystem-driven permafrost in the boreal region.

We differentiated three quasi-stable permafrost conditions and four degradation stages using a system modified from Jorgenson (2021) to address the complicated nature of permafrost formation and degradation in boreal ecosystem-driven

permafrost (Supplemental Information Table 1; Supplemental Information Fig. 5). For permafrost types, undegraded (UD) was assigned when thaw depths were less than 0.8 m (typical maximum range of late summer thaw for organic-rich soil in the Fairbanks area; Douglas et al., 2021). We considered this "stable" permafrost. Multi-year frost (MF) was thin frost (typically 10-30 cm thick as detected by probing through the frozen layer) that persisted for one to several years. New repeat permafrost (NPR) had a frozen layer from 1 to several meters thick (detected through coring) that formed in old thermokarst bogs and persisted for decades or more. For degradation stages, degradation-initial (DI) occurred where thaw depths increased over time to ~1.1 m depths. Degradation-progressive-shallow (DPS) was assigned when the permafrost surface was detected below 1.1 m indicating development of an open talik (unfrozen zone between seasonal frost and permafrost). Degradation-progressive-deep (DPD) identified permafrost that retreated below probing depth (>2.5-3 m). We grouped these three into a broader category of vertical degradation that we considered "unstable" permafrost. Degradation-lateral (DL) was used where permafrost thawed along margins of permafrost plateaus; these areas quickly joined "through" taliks (thawed zone penetrating all the way through permafrost) of adjacent old thermokarst bogs or fens. A final degradation stage ("degradation complete"), which is not plotted, represents "old thaw" which occurs in locations within old thermokarst bogs and fens where permafrost was not detected within probing depth and where we assume a through talik had developed.

To independently estimate top-down and lateral permafrost thaw based on repeat thaw probing we stratified each transect into three zones: (1) "old thaw" which were completely thawed in 2012 that we associated with thermokarst bogs and fens; (2) "lateral thaw" where permafrost thawed along margins of permafrost plateaus between 2012 and 2020 with thaw extending below probing depth (2.5-4.0 m depending on rod extensions and occurrence of gravel); and (3) "vertical thaw" of top-down permafrost where near-surface permafrost was detected with the 3 m probe.

Within vertical thaw zones we further differentiated two surface conditions useful for assessing ERT measurements. First, "stable" permafrost where thaw depths were always less than 0.8 m, typically within the range of annual variability, in organic-rich, fine-grained soils. Second, "unstable" permafrost where thaw depths exceeded the normal range or where an unfrozen region (open talik) developed from 2012 to 2020 between the bottom of the active layer and the top of near-surface permafrost.

For ERT measurements we used an Advanced Geosciences Incorporated (Austin, Texas) SuperSting R8 eight-channel portable meter with 84 electrodes at 2 m spacing for transects T1, TF88, and TF01 and 1-m spacing for transect TF10. Electrodes ranged from 0.45 m to 1 m in length. More detailed operational information for measurements made in 2012 is provided in Douglas et al. (2016). A dipole-dipole measurement geometry was used for all ERT surveys due to its sensitivity to lateral features and our focus on changes in near-surface permafrost distribution (Douglas et al. 2016; Minsley et al., 2022).

ERT survey data were processed with the R2 family of codes using open-source ResIPy (Blanchy et al., 2020) software to remove noisy data points (i.e. resistivity values with noise greater than 3%) and find inverse model solutions using least-squares inversion methods (Loke and Barker, 1996; Loke et al., 2003). Approximating resistivity differences between 2012 and 2020 measurements was achieved by projecting 2020 resistivity measurements onto the 2012 inversion mesh and calculating the log difference within each mesh element relative to the 2012 inverse model solution according to:

$$\Delta\Omega m_{2012-2020} = log(\Omega m_{2012}) - log(\Omega m_{2020}).$$

Here, $\Delta\Omega m_{2012-2020}$ represents the log difference in inverse model solutions between 2012 and 2020 field campaigns. We acknowledge topographic differencing errors from performing an inversion of 2020 resistivity data on the 2012 inversion mesh and incorporate differencing errors into our final interpretations and discussion of uncertainty in the following sections.

### 2.2.2 Airborne LiDAR data acquisition and analysis

Airborne LiDAR data was first acquired over the study area in May 2014 by Quantum Spatial Incorporated (Anchorage, Alaska). Detailed information on operating conditions and post processing is provided in Douglas et al. (2016). A second acquisition was made by the same vendor in May 2020. Both datasets were collected at a high point density (larger than 25 points/m$^2$) in a period of 95% snowmelt with leaf-off conditions. From this very fine resolution LiDAR DEM products were generated at 0.25 m resolution. A total of 183 Ground Control Points (GCPs) were measured using real time kinematic and post processed kinematic techniques to validate the accuracy of the LiDAR DEM products. The average vertical elevation error was 0.093 m in the DEM (95[th] percentile) across different ecotypes (Zhang et al., 2023), leading to the propagated error of 0.13 m for the differenced LiDAR DEM. This error was considered in our thaw stage analysis. We extracted LiDAR DEMs and calculated the elevation difference between 2014 and 2020 in a 300 m by 500 m buffer strip encompassing each transect to map thaw degradation stages and quantify thaw extent and volumetric loss within the buffered transect areas.

From repeat LiDAR DEMs and corresponding elevation changes we mapped the three thaw degradation types along each transect to be consistent with our field stratified zones: (1) old thaw as areas that had minimal elevation change within the thermokarst bogs and fens which often had a low elevation within a landscape; (2) lateral thaw as areas adjacent to old thaw areas that collapsed to the same elevation as the old thaw areas and had a larger elevation change; and (3) vertical thaw areas as the remaining terrain which was often found at permafrost plateaus covered by trees.

We identified elevation change and DEM thresholds for each zone based on field measurements which were then applied to map thaw degradation stages to quantify top-down thaw, lateral thaw, and corresponding volumetric changes for each buffered transect using repeat LiDAR DEMs. To map thaw degradation stages, we applied an object-based analysis approach in which a multiresolution segmentation was used to segment the LiDAR DEM difference image first, and then the mean DEM of 2014 and 2020 and the DEM difference were calculated. This object-based analysis approach reduces "salt-and-pepper" effects which are caused by isolated pixels with high spatial heterogeneity. This abnormality is considered as noise affecting analysis accuracy and results (Blaschke et al. 2000). Thaw extent and volumetric loss from lateral and vertical thaw were then calculated based on corresponding area and elevation changes for each object and summarized for each buffered transect.

# 3 Results

## 3.1 Degradation stages identified along the transects

We compiled the repeat surveys of surface topography, water depth, thaw depth, and ancillary subsurface measurements from the four transects to define trends of subsurface thaw and corresponding ground surface settlement between 2012 and 2020 (Figs. 2 and 3). Dates when measurements were collected are included in these Figures. We differentiated trends across each transect into six degradation/aggradation stages (Supplemental Information Table 1). UD (thaw depths remained <100 cm with annual changes <30 cm) occurred along 29% of the transects in 2012 and decreased to 8% by 2020 (Figs. 2 and 3). DI (thaw depths ~ 1.1 m or had increased by >30 cm as a brief transitional stage) occurred along 36% of the transects. DPS (vertically increasing thaw depths >120 cm but not more than ~250-300 cm), increased from 12% to 20% between 2012 and 2020. DPD (thaw depths increased to >250-300 cm) increased from 6 to 16%. Together, these shallow and deep progressive degradation stages indicating top-down thaw of near-surface permafrost increased from 18% to 36% over the 8 year study period. We presume DPD denotes areas with open taliks. DL (lateral thaw bogs and fen margins) increased slightly from 3% to 6%. Regions characterized as DCO occurred under old bogs and fens and presumably had completed degradation through the entire permafrost zone to form through taliks. They increased slightly between 2012 and 2020 from 35% to 44%. We attribute this change, however, due to loss of Repeat-Permafrost-Thin (RPT) and MF (Figs. 2 and 3) near the surface. MF was used to differentiate areas with a thin layer (typically <30 cm) of frozen ground that persisted for 14 years (permafrost is conventionally defined as frost persisting more than 2 years). MF decreased from 7% to 0%.

Overall, degradation changes were mostly minor from 2012 to 2014 but exhibited large changes by 2020 (Fig. 4). We highlighted differences in degradation stages across the four transects using observations along the first 200 m where measurements were consistently taken over time. Trends were highly variable among transects in response to differences in soil texture, ground-ice content, surface organics, old thermokarst history, and fire history.

UD showed steep losses across all transects (0-200 m), and was eliminated at TF01 and TF10. TF88 had the highest initial percentage (59%), which persisted from 2012 to 2014, but then dropped to 27% by 2020. The more recently burned TF01 and TF10 had the steepest losses during 2012-2014. DI showed a substantial increase at TF88 by 2020 (20%) and a substantial decrease at TF10 as the stage transitioned to DPS.

Degradation-progressive, both shallow (DPS) and deep (DPD), fluctuated substantially across the four sites, in large part due to the transition of the shallow phase into the deep phase. TF10 was the only transect to show a large, consistent increase in DPS (from 8% to 48%). DPD showed a large increase at TF01 (38%), small increases along T1 and TF88 (7%), and no change at TF10. We attribute the large increase in DPS at TF10 to the short-term response to the 2010 fire and the large increase in DPD at TF10 to a longer-term response to the 2001 fire.

DCO was abundant across all transects; it held fairly steady at TF01 and TF88 but showed large increases at T1 and TF88 due to the loss of RPT and MF near the surface above the unfrozen zone at depth. RPT was used to differentiate young bodies of permafrost formed in response to ecological succession and periods of extremely cold and dry winters (Jorgenson et al.

2020). RPT occurred only within an old bog at TF88 (Figs. 2 and 3) and was measured to be 193 cm deep based on core TF88-060-2012. We estimated it formed during extremely cold winters in the 1960s; it was last observed in 2014 and disappeared by 2020. MF was evident at all transects during 2012-2014 and had completely disappeared by 2020. MF was most prevalent at T1 (23%) near the surface of the old thermokarst bog. When this thawed the segments reverted to DCO. At the other transects, MF covered less than 4%.

### 3.2 Thaw settlement revealed from field measured topographic data

Thaw settlement varied nearly ten-fold across degradation stages and transects based on differences in relative ground heights of degradation stages in 2020 relative to UD in 2012 (Figs. 2 and 3). Overall, mean thaw settlement was deepest for DL compared to other stages at TF01 and TF88 while DPD was deepest at T1 and DPS was deepest at TF10. DL was 918 cm

deeper than DCO, indicating that over time accumulation of post-thermokarst peat raised the ground surface. Overall, thaw settlement across all stages tended to be deepest at TF88 and T1, and shallowest at TF01 and TF10. We attribute the high variability to differences in soil properties, ground ice contents, and the timing of degradation onset.

The ratio of increased ground surface thaw settlement to thaw depth (Fig. 5) directly relates to excess ice content. For example, a ratio of 0.38 indicates an average excess ice volumetric content of 38% in the permafrost profile before thawing.

Overall, the relationship was moderately strong ($R^2$=0.61, n=239), but varied slightly among transects ($R^2$ ranged from 0.51 to 0.70). The ratio increased nearly three-fold between 2012 and 2020 between TF01 (0.23 regression slope coefficient) to T1 (0.63). The overall ratio for all measurements was 0.38. Scatter in the relationship indicates substantial variability which we attribute to differences in soil characteristics and surface conditions. For example, high thaw depths with negligible settlement occurred in sandy soils along TF10. Floating mats along laterally collapsing margins also were a problem, especially along

TF88 where floating mats developed over time raising the surface even while thaw depths continued to increase. Without this influence, the relationships would have been stronger and the ratio would be larger.

### 3.3. Mapping thaw degradation stages from repeat LiDAR

Maps of LiDAR DEM differences between 2014 and 2020 (Fig. 6) show ground surface elevation changes over the 6-

285  year period. In our transect surveys we did not identify any areas that rose in elevation over time, so we attribute increases in elevation (positive values in Fig. 6) to higher vegetation or higher water levels in 2020. Lateral thaw into the permafrost plateau is evident at all four sites. TF10 shows the greatest subsidence of the plateaus themselves.

To map our three defined thaw degradation zones (old thaw, lateral thaw, and vertical thaw) elevation thresholds in 2014 and thresholds of elevation change between 2014 and 2020 were identified from the field surveyed ground elevation, thaw

depth data, and other ancillary data we collected along each transect. For the unburned site T1 old thaw was found at fens and bogs with elevation less than 128.24 m and an elevation change less than 0.1 m. For TF01, the identified elevation threshold

was 123.23 m in 2014 and the elevation difference threshold was 0.15 m; for TF10, an old thaw was found at elevation less than 127.49 m with elevation change less than 0.15 m; for TF88, old thaw was found at elevation less than 131.79 m with elevation change less than 0.1 m. Based on these thresholds, old thaw areas were objects with an elevation in 2014 less than the identified threshold and elevation change less than the threshold 0.1 m for T1 and TF88 and 0.15 for TF01 and TF10. Once the objects with old thaw were identified, these objects were spatially dissolved, and adjacent unidentified objects would be the potential objects as lateral thaw. These adjacent objects were further refined and finalized as lateral thaw areas if they had larger elevation change due to ground surface collapse (larger than the threshold). The remaining objects were labeled as vertical thaw.

Fig. 7 shows thaw degradation zones classified by these rules. The maps delineate well the thaw extent and stages for old thaw found over bogs and fens (in green) where DCO was observed. Lateral thaw degradation (in blue) is evident along the edges of bogs and fens. Vertical thaw (in red) over high elevation permafrost plateau areas denotes where top-down thaw occurred due to climate warming or fire disturbance.

The estimated thaw extent and volumetric loss from lateral and vertical thaw for each buffered transect is displayed in Fig. 8 and summarized in Supplemental Table 2. TF10, the 2010 fire scar, had the largest lateral thaw of 26,555 m$^2$ (17.7%) and corresponding volumetric loss of 5,015 m$^3$. Its vertical thaw was also relatively high with an extent of 69,125 m$^2$ (46.1%), leading to the largest volumetric loss of 9,755 m$^3$ among all transects. The largest extent of vertical thaw was observed at TF88 (91, 206 m$^2$, 60.8%) but the volumetric loss due to vertical thaw was moderate (3, 350 m$^3$).

## 3.4. Electrical resistivity tomography measurements

ERT measurements were made across the transects in 2012 and 2020. Details from the 2012 collection and relationships between fire disturbance and permafrost degradation along the transects between 2012 and 2014 are presented in Douglas et al. (2016). Similar operating conditions were repeated for the 2020 collection (Fig. 9). This allowed us to quantify the log difference in resistivity values at depths up to 20 m over the eight years of elapsed time.

Resistivity values greater than 600-800 Ωm (log$_{10}$ resistivity values of 2.8-2.9) have been found to correlate with syngenetic permafrost in the greater Fairbanks region (Hoekstra and McNeill, 1973; Douglas et al., 2008). As such, values above this are generally associated with frozen ground. Values below log$_{10}$ resistivity of 2.8 Ωm are considered absent of permafrost and we interpret those areas as unfrozen. In both 2012 and 2020 the major unfrozen zones and permafrost plateaus are readily apparent from their higher resistivity values and low active layer depths. Resistivity values decreased between 2012 and 2020 and these correspond with the thaw degradation stages identified earlier.

All transects had areas exhibiting decreases in resistivity values and associated ground surface subsidence over the eight-year study period indicating top-down thaw. TF10, the transect representing the most recent wildfire, yielded the greatest values for top-down thaw (up to 3 m) and associated subsidence (~1 m). Permafrost plateaus present at TF01 (-20 to 25 m and 50 to 95 m) and TF10 (0 to 20 m; 40 to 80 m) in 2012 experienced major subsidence by 2020. Permafrost plateaus at TF88

(50 to 70 m; 125 to 180 m) also showed thaw subsidence by 2020. The site with the second greatest top-down thaw values (~ 2 m) was the second most recent fire (TF01). Thawed areas correspond with large percent decreases in resistivity values over time.

## 4. Discussion

Permafrost in Interior Alaska has been slowly thawing for the past ~500 years with sporadic periods of accelerated thaw
typically attributed to wildfire and subsequent permafrost stabilization or aggradation associated with forest succession (Jorgenson et al. 2001; Jones et al., 2013). Air temperature increases since the 1970s in Interior Alaska (Osterkamp, 2005) and across the Arctic (Smith et al., 2022) have led to increased permafrost temperatures and widespread thaw. Numerous recent studies in Interior Alaska show an acceleration of permafrost degradation with deeper seasonal thaw depths (Douglas et al., 2020; Euskirchen et al., 2024), widespread talik expansion (Farquharson et al., 2022), and an increased thermokarst
development (Douglas et al., 2021; Minsley et al., 2022; Brodylo et al., 2024). Studies from sites across the Arctic show increasing soil temperatures (Chen et al., 2022) and subsidence due to permafrost degradation (Streletskiy et al., 2024).

All four transects contain low lying old thermokarst bogs surrounded by permafrost plateaus consisting of birch or aspen forests, black or white spruce woodland, or mixed deciduous and conifer forest (Figs. 2 and 3). Plateaus are the most common landforms occurring in boreal discontinuous permafrost, and are a signature indicator of the presence of
340 permafrost in the region. At T1, an unburned site, the top of near-surface permafrost below the collapsing birch forest increased steadily from ~0.5 m in 1999 to ~2 m in 2020. TF88, which burned in 1950 and 1988, also shows top-down thaw associated with areas of collapsing birch forest but not as much subsidence (~0.5-1 m) associated with this thaw as T1. This is likely due to the lower ice content of lenticular ice from 1.2 to 2.8 m depth at TF88 compared to braided (reticulate-platy) ice from 1.5 to 2 m at T1 (Brown et al., 2015).. TF01, a fire scar from 2001 dominated by burned black
spruce, was already showing signs of top-down thaw. For example, the top of near-surface permafrost was ~2 m deep in 2011 and it increased to >3 m at some locations during our study period. However, subsidence associated with this thaw was lower than at other transects, likely due to the lowest ice contents in the upper 3 m of any of the four sites (Brown et al., 2015). Permafrost warming and related top-down and lateral thaw continued between 2012 and 2020. TF10, the most recent fire scar, had the highest rates of top-down thaw (>2 m) and related surface subsidence (up to 1.2 m) of any site. These results support
numerous studies showing that the first decade following a wildfire is typically associated with the most rapid surface warming and that permafrost thaw slows after vegetation starts to recover (Gibson et al., 2018; Holloway et al., 2020).

### 4.1 Relative strengths and weaknesses of different permafrost degradation measurements

No single method provides all the information typically needed to adequately assess permafrost undergoing degradation or aggradation. For example, frost probing yields insight into top-down thaw or indicates areas where near-surface permafrost

is aggrading upward but with limited spot measurements. LiDAR allows the identification of vertical and lateral subsidence upon thaw or heave associated with aggradation at a relatively larger spatial coverage but is limited to surface measurements. ERT can identify the presence/absence of permafrost at tens of meters depth but is not as well suited as frost probing for survey-level measurements of vertical or lateral degradation of permafrost bodies.. A combined use of three methods is more effective to characterize permafrost degradation at multiple dimensions than the application of each individual method. Though our study focused on sites in Interior Alaska the methods we applied here can be used to survey and track changes in other permafrost terrains.

### 4.1.1 Repeat thaw probing and ground surveys

Thaw probing, accompanied by surveying of ground and water-surface elevations, yielded the most precise measurements of subsurface changes in the permafrost table than ERT or repeat LiDAR and this helped identify degradation stages (Figs. 2 and 3). Probing can detect changes in soil texture (peat, silt, sand, gravel) with depth, however, soil stratigraphy was confirmed with soil pits and boreholes (Brown et al., 2015). It is important to note the importance of measuring both thaw depth and thaw settlement, which allows calculation of ice content and accurate measurement of the change in the elevation of near-surface permafrost. Ice content measurements, where available, can be used to compare to estimated ice contents.

For the most part the top of near-surface permafrost measured from frost probing (< 3 m) matches the presence of permafrost inferred from sudden increases in resistivity to values ~2.8 $\Omega$m. At all sites the margins between bog features and permafrost plateaus degraded laterally during the study period. Douglas et al. (2016) showed irregular margins along the edges of the permafrost bodies at the four transects. Tabular "shelves" were evident in ERT measurements, particularly at T1 ("1930" in the earlier study), TF01, and TF10. Other studies of permafrost thaw and thermokarst development in peatlands have shown lateral thaw degradation created thermal "niches" of higher subsurface thaw rates underneath "shelves" (Jorgenson et al., 2012; O'Donnell et al., 2012). These features can lead to thaw probe measurements of near-surface permafrost and aerial photo imagery of landforms associated with permafrost where tabular permafrost areas are only a meter or two thick. As such, it is likely the amount of permafrost is overestimated in these areas. Our repeated geophysical and thaw probe measurements show these tabular bodies can thaw completely in just 8 years. It likely takes longer for vegetation to respond to the changes in ground surface conditions associated with permafrost loss, for example, for trees to get waterlogged and die and for vegetation associated with standing water to colonize those areas.

The large disadvantage of probing is that it is limited to the top 2-3 m of fine-grained soils and determining frost boundaries becomes more uncertain (10 to 30 cm error) as unfrozen water contents increase during partial permafrost thaw. At depths greater than 3 m, determination of penetration refusal due to frost becomes less reliable due to friction along the probe. Also, the occurrence of sand and gravel at depth impedes probing. Borehole logging of soils can reduce some of these uncertainties.

### 4.1.2 Repeat airborne LiDAR

Repeat LiDAR (Figs. 6 and 7) provides high-precision measurements of surface-elevation change, and can be used to model vertical versus lateral thaw (thaw along changing permafrost margins) across large areal extents. Collections in 2014 and 2020 showed that almost every permafrost plateau margin shrunk as lateral thaw expanded old thermokarst bogs into plateaus. All of the plateaus exhibited vertical (top-down) thaw of near-surface permafrost. No areas at any of our buffered transects exhibited signs of permafrost aggradation or increases in ground surface elevations that are not attributable to higher vegetation or elevated water levels in 2020 compared to 2014.

Since the transects contain different aerial extents of old thermokarst bog and permafrost plateau terrains we cannot compare the percentage of lateral and vertical thaw across the transects (Supplemental Table 2). However, in all four buffered transect areas vertical thaw extents and volumetric loss were greater than lateral thaw. The most recent fire scars (TF01 and TF10) yielded the greatest rates of vertical and lateral thaw extent and volume loss. TF10 exhibited more than twice the amount of thaw as TF01 and this supports ground surface measurements showing the greatest thaw subsidence across transect TF10 (Fig. 3). As stated earlier, differences in rates and volumes of vertical and lateral thaw at a given transect are likely attributable to the different ice contents of the near-surface permafrost that thawed.

Changes in surface water conditions, vegetation growth, and floating soil mats in thermokarst features affect LiDAR surface elevation measurements. Also, airborne LiDAR does not provide information about below ground soil or permafrost characteristics. We used settlement of >0.3 m as the threshold for detecting significant permafrost thaw, but it is notable that substantial areas of old thermokarst bogs and fens also had water-level changes greater than that. Thus, repeat airborne LiDAR is more reliable in ice-rich areas (T1 and TF88) than ice-poor ones (TF01 and TF10). In addition, LiDAR can detect permafrost thaw only where the ground surface remains above water. If the original ground surface settles below the water level or precipitation increases markedly from one acquisition to the other, thaw settlement can be masked by surface water. For example, the summer of 2014 had more total wet precipitation than the summer of 2020 but far more rain fell from late July through September in 2020 than in 2014. Hence the water table elevations of bogs were higher in 2020 than in 2014 (Fig. 6). This elevated water table masks the water feature boundaries that were present in 2014.

### 4.1.3 Repeat Electrical resistivity tomography

Repeat ERT (Fig. 9) has a strong advantage in delineating the upper boundary of near-surface permafrost down to depths of 20 m or more. As such, this method is particularly useful at identifying top-down degradation and zones of deep progressive thaw, however, vertical changes in permafrost extent or stratigraphy should be confirmed with boreholes where possible. In interior Alaska resistivity values above 600-800 $\Omega$m ($\log_{10}$ resistivity values of 2.8 to 2.9 $\Omega$m) correlate with conductive permafrost material (Hoekstra and McNeill, 1973; Douglas et al., 2008; 2016; Minsley et al., 2022). At our sites the top of near-surface permafrost corresponds with sudden decreases in $\log_{10}$ resistivity from values greater than 2.5 $\Omega$m above the

permafrost (active layer) to values between 3 and 4 Ωm where permafrost is present. Old thermokarst bogs are readily identified by low resistivity values and permafrost plateaus have rapid vertical changes in resistivity over the upper ~3 m that are confirmed as permafrost by thaw probing. Resistivity values decreased between 2012 and 2020 for every measurement location across all transects at depths of up to ~20 m. The difference map between the two resistivity campaigns shows the greatest decreases in resistivity occurred in permafrost plateaus and this supports surface surveys along the transects (Figs. 2 and 3).

ERT can identify subsurface massive ice bodies (Herring et al., 2023) and is sensitive to changes in unfrozen-water content that accompanies changing permafrost temperatures. However, ERT is less precise in delineating lateral changes in permafrost bodies, presumably in part due to its sensitivity to unfrozen-water content. Both thaw probing and ground-based ERT are time intensive with limited coverage, and require movement across the landscape. Various ERT measurement sequences—i.e., permutations of current-injecting and potential electrode pairs—present tradeoffs in lateral versus vertical sensitivity and data acquisition efficiency (Binley and Kemna, 2005). The dipole–dipole configuration employed in this study is particularly sensitive to lateral resistivity contrasts and near-surface features, offering the advantage of relatively rapid acquisition times (Oldenburg and Li, 1999) and validation with thaw probe measurements. While model solutions derived from our ERT data suggest permafrost at depths exceeding 20 m (Fig. 9), the validity of such features remains uncertain in the absence of borehole corroboration. Given the known limitations of dipole–dipole geometries at depth, compounded by conductive near-surface layers that attenuate sensitivity with depth, we refrain from interpreting the lower boundary of permafrost in this dataset. Instead, we emphasize that strong confidence exists in detecting top-down thaw and near-surface changes, which align with ground-based probing across all transects.

## 4.2 Degradation and aggradation stages

Permafrost thaw can occur on the surface, laterally along the margins, upward from the bottom, and through interior channels. Categorizing degradation and aggradation stages is helpful in assessing permafrost dynamics (Fig. 4). We differentiated five degradation stages that improved our ability to detect permafrost loss. The initial degradation stage differentiates the initial increase in the active layer before talik development. Two of the older burned areas (T1 and TF88) showed increases in initial degradation in response to recent climate warming, while the more recently burned areas (TF01 and TF10) showed decreasing initial degradation with transition rapidly into shallow and deep progressive degradation associated with open talik development. Complete degradation in old thermokarst bogs and fens was widespread across all transects indicating permafrost degradation has been occurring for centuries (Kanevskiy et al. 2014). This differentiation allowed us to distinguish new from old thaw and the increasingly widespread onset of initial thaw.

We found several ages of permafrost that formed in response to climatic change and ecosystem development. At TF10 and T1 some permafrost soils lacked plant macrofossil evidence of previous thermokarst indicating permafrost formed during the early-mid Holocene (Jorgenson et al., 2001; Brown et al., 2015). At TF88 we found permafrost 23 m thick that had formed low mounds (~0.3 m) in the central portion of an old thermokarst bog, presumably formed during extremely cold winters in

the mid-1960s. These mounds disappeared by 2020. At T1 we documented thin (0.1-0.3 m) multi-year frost in soils above shallow taliks and in old thermokarst bogs, mostly during the early 2000s. This sporadic occurrence of multi-year frost greatly complicates the use of mean thaw depths as an overall metric for monitoring permafrost degradation. We presume multi-year frost will no longer form under the warming climate.

## 4.3 Trends in lateral versus vertical thaw

We found only small changes in the extent of lateral thaw and sharply increasing vertical thaw (initial and progressive degradation combined) from 2012 to 2020, indicating a shift toward widespread vertical thaw and talik development. From 2010 to 2012 the transects had mostly undegraded permafrost on permafrost plateaus and completely degraded permafrost in
old thermokarst bogs and fens. By 2020, undegraded permafrost had mostly disappeared.

During colder climate periods, particularly during the Little Ice Age, we think lateral degradation was the primary mechanism for permafrost thaw in boreal regions because of the strong effects of surface and groundwater (Jorgenson et al., 2010; 2022). Recent warming during the last decade, however, has brought near-surface permafrost temperatures close to 0 °C (Smith et al., 2022; Jorgenson et al. in press) and led to recent talik development (Farquharson et al., 2022). In contrast,
Jorgenson et al. (2020) found mean lateral degradation rates for permafrost along fens changed little from 1949 to 2018. Furthermore, thermokarst fen and bog expansion in terms of area has been relatively slow: thermokarst fens increased from 3.1% to 4.3% and thermokarst bogs increased from 0.8% to 2.0% from ~1949 to ~2017 (Jorgenson et al. 2020).

## 4.4 Factors affecting degradation and aggradation

Monitoring of lateral and vertical thaw across our four study areas has revealed climate, surface and groundwater, soil characteristics, fire, and vegetation-soil-water interactions all affect rates and patterns of permafrost degradation. Mean annual temperatures in Alaska have increased nearly 2 °C since the 1920s (Douglas et al., 2025), with eight of the ten warmest years occurring since 2005 (Walsh et al., 2020). Annual precipitation increased 9% since 1970 with freezing rain events nearly 3 times higher in the 2010s compared to earlier periods (Thoman and Walsh, 2019). Notable extreme precipitation events have
occurred in the 2010s and are linked to accelerated permafrost degradation (Douglas et al. 2020). Warmer, snowier winters, which reduce wintertime permafrost cooling, have become more prevalent since 2005 (Jorgenson et al., 2020).

Surface and groundwater temperatures have a large effect on permafrost degradation. Mean annual water temperatures at the bottom of shallow ponds can be 9-11 °C higher than adjacent deep soil and, thus, historically has been a main driver of lateral degradation and thermokarst expansion around lakes and bogs (Jorgenson et al., 2010). Mean annual groundwater
temperatures in thermokarst fens (~1. 2 m depth) on the Tanana Flats were frequently near 3 °C, with above freezing temperatures persisting through winter. All study areas had widespread thermokarst bog coverage with extensive lateral thaw while only two transects (T1 and TF88) had broad thermokarst fen coverage associated with groundwater movement. Transect

T1 was unique in having large fens on both sides of a long, narrow permafrost plateau, and had rapid degradation and collapsing birch forests along the portion of the transect adjacent to the fen. This transect had the largest loss of permafrost despite it being unburned.

Soil texture and ground ice contents are highly variable across Tanana Flats (Jorgenson et al., 2001; Brown et al., 2015). This has a large effect on degradation patterns and rates and our ability to detect changes. The sandier soils and lower ice contents at TF01 and TF10 contributed to more rapid thaw and lower thaw settlement than at the more ice rich and peaty transects T1 and TF88. Sand and gravel at depth in fluvial deposits on abandoned floodplains likely contribute to rapid subsurface thawing as groundwater penetrates deeper permafrost. Thus, varying depositional patterns during floodplain development left a legacy of soil and ground ice conditions that can greatly affect permafrost aggradation and degradation (Kreig and Reger 1982; Jorgenson et al. 2022).

Fire can dramatically increase permafrost thaw by removing the vegetation canopy, destroying insulating soil organic material, and changing surface albedo (Brown et al., 2015; Holloway et al., 2020). Differences in soil and hydrology described above confounded our ability to rigorously analyze fire effects. However, large differences among sites allow us to highlight broad effects. First, at the transect with the most recent wildfire (TF10) there was a nearly immediate and rapid increase in shallow progressive degradation following the fire. It was the first time we have observed a fire-induced rapid collapse of permafrost terrain to below water level. Second, both of the more recently burned transects (TF01 and TF10) rapidly lost all undegraded permafrost during the early 2010s during a period of substantial climate warming, compared to the older transects (TF88 burned in 1950 and 1988, and T1 unburned since the 1930s) where undegraded permafrost was more persistent. At odds with these trends, the unburned transect T1 also had rapid loss of permafrost due to its proximity to large groundwater fens, and transect TF88, which burned twice, had the largest percentage of undegraded permafrost.

Finally, as permafrost thaws, changes in vegetation, soils, and surface water interact to provide strong feedbacks that affect both permafrost aggradation and degradation. Initially, during collapse of permafrost terrain in flat lowlands, water impounds at the surface providing a positive feedback facilitating further thaw-induced lateral expansion (Westermann et al., 2016; O'Neil et al., 2023). This lateral degradation was prominent in all study areas. Quickly, however, aquatic sedges, forbs, and mosses colonize the flooded ground and initiate a successional sequence that contributes organic matter for peat accumulation and lifts the surface above the water level. These older portions of bogs and fens were prevalent at all transects. At transect TF88 this successional development contributed to the repeated thin permafrost formed within an old bog that was probably initiated during the colder climate of the 1960s. At transect T1, very thin multi-year frost was repeatedly observed in the central old thermokarst bog.

## 5. Conclusions

It is clear from our repeat ground surface surveys that interior Alaska permafrost is undergoing widespread and dramatic thaw degradation. For example, at site T1, our control transect, top-down thaw of near surface permafrost occurred over 44% of our study sites between 2004 and 2020. Multi-year frost and repeat thin permafrost, two types of permafrost aggradation,

were almost completely absent by 2020. To better assess these changes in permafrost at our sites we differentiated degradation stages with an emphasis on contrasting vertical versus lateral thaw. Shallow and deep progressive degradation (i.e. top-down thaw of near surface permafrost) increased from 18% to 36% across our study sites over the 8 year study period. Lateral thaw of tabular shaped permafrost boundaries and development of unfrozen zones between the bottom of the seasonally frozen layer and the top of near-surface permafrost (taliks) were evident, supporting recent identification of widespread talik development across the region (Farquharson et al., 2022). Repeat airborne LiDAR also identified lateral and vertical loss of near-surface permafrost at all our sites with 60% of the area at TF88 site exhibiting vertical thaw (Fig. 8). Much of the permafrost thaw at our sites is associated with the press disturbance of climate warming (Douglas et al., 2021; Farquharson et al., 2022). However, rapid loss of near-surface permafrost was initiated immediately after the pulse disturbance of the 2010 fire (TF10; Fig. 3) with subsidence of up to 1 m in the subsequent decade.

There are numerous approaches for assessing permafrost changes, including thaw probing (Nelson et al., 2021; Douglas et al., 2020), temperature monitoring (Farquharson et al., 2022), ERT (Douglas et al., 2016; Minsley et al., 2022; Herring et al., 2023), LiDAR measurements (Douglas et al., 2022), photogrammetry (Van der Sluijs et al., 2018), and synthetic aperture radar (Zweiback and Meyer, 2021), and remote sensing of thermokarst landforms (Jorgenson et al., 2016, Kokelj et al., 2020). We used results from repeated thaw probe measurements, ERT, and airborne LiDAR to assess their relative strengths and weaknesses for quantifying changes in ecosystem-driven discontinuous permafrost in a boreal ecosystem. To better assess these changes, we differentiated degradation stages with an emphasis on contrasting lateral versus vertical thaw. Across all methods, we found monitoring boreal permafrost is challenging because the three-dimensional nature of permafrost bodies makes them sensitive to climate, hydrology, water-soil-vegetation interactions, and fire disturbance. Our assessment of the three disparate but commonly used methods to assess changes in permafrost extent over time identifies advantages and disadvantages of each method.

The interactions among climate, hydrology, soils, vegetation, and fire disturbance create a highly patchy mosaic of stable and degrading permafrost. It also makes for a challenging environment to detect permafrost degradation vertically and laterally or where permafrost and thermokarst are of various ages and in differing stages of formation and aggradation. Our monitoring, however, came at a time of recent substantial increased temperatures and precipitation, and we believe we have detected a large shift from past narrower and slower lateral degradation to more rapid and widespread vertical degradation. Given the positive and negative aspects of ground surface surveys, airborne LiDAR, and geophysical investigations a coupled application of these methods is warranted to track permafrost thaw at similar locations or other permafrost regions. Future applications of these methods should apply geospatial analyses and modern artificial intelligence to identify key variables relating surface and subsurface conditions to project finer scale field-based spatial assessments across broader regions. Of particular interest are ways to combine these techniques with multiple airborne and spaceborne remote sensing products to identify relationships of key ground state variables that control permafrost stability like the depth of the snowpack, surface soil moisture, soil strength, surface water ponding, and seasonal subsidence.

**Data availability statement**

The data that support the findings of this study are available upon reasonable request from the authors.

**Author Contribution**

TAD oversaw study activities. MTJ designed and led field measurements. TDS applied geophysical methods and analyses. CZ
developed and applied geospatial analyses of thaw and generated map products. All authors contributed to data analysis and
manuscript preparation.

**Competing Interests**

The authors declare that they have no conflict of interest.

**Acknowledgements**

We appreciate the efforts of numerous amazing scientists and engineers that participated in field measurements over the years
as well as the safe and competent job done by many helicopter pilots. We thank multiple peer reviewers and the Editor for
their suggestions on improving the manuscript.

**Financial Support**

This research was funded by the Department of Defense's Strategic Environmental Research and Development Program
(Project RC18- 1170) and Environmental Science and Technology Certification Program (Project NH22- 7408) as well as the
U.S. Army Engineer Research and Development Center Basic Research Portfolio through Program Element PE
0601102A/T14/ST1409.

**ORCID iD**

Thomas A Douglas https://orcid.org/0000-0003-1314-1905

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

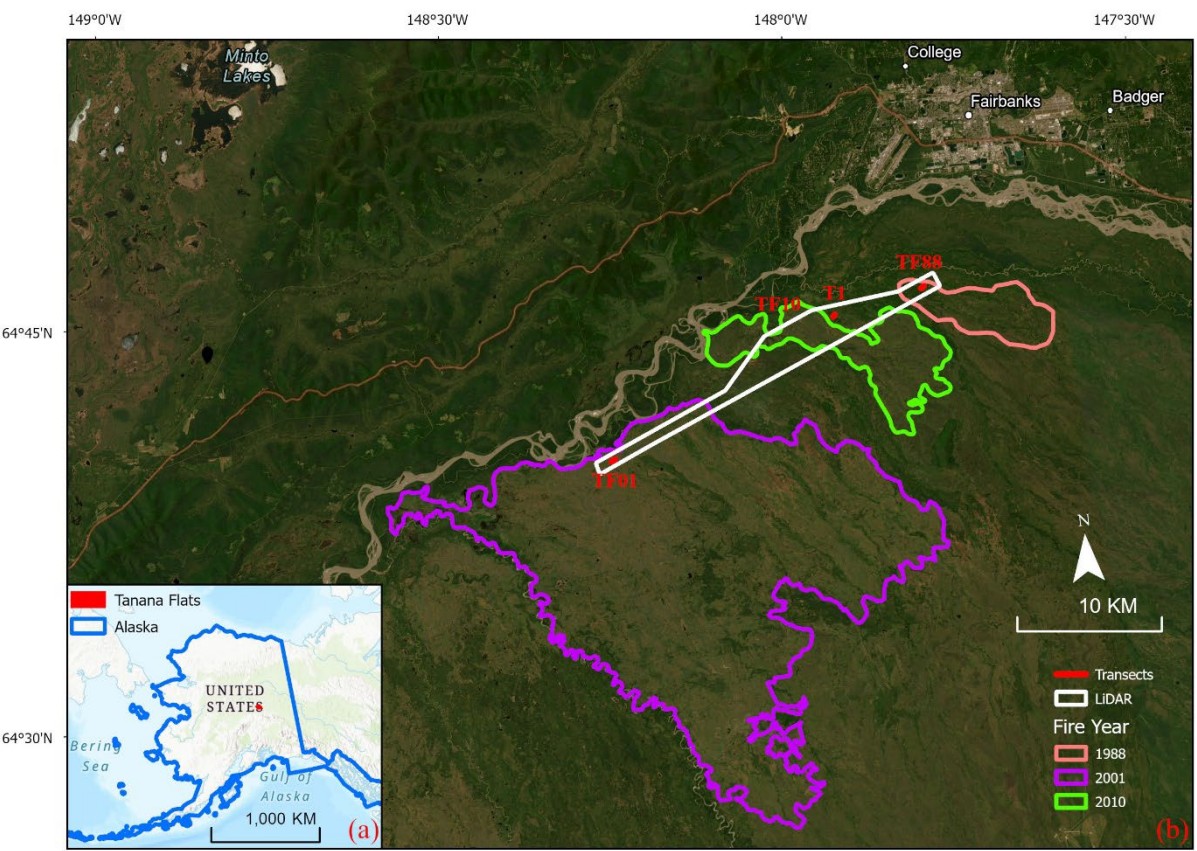

**Figure 1. Study area in Interior Alaska (a) identifying the regional location in Alaska and (b) the location of four transects: T1 (unburned), TF88 (burned in 1988), TF01 (burned in 2001), and TF10 (burned in 2010). Perimeters for the 1988, 2001, and 2010 fires are also provided from the Alaska Interagency Coordination Center**

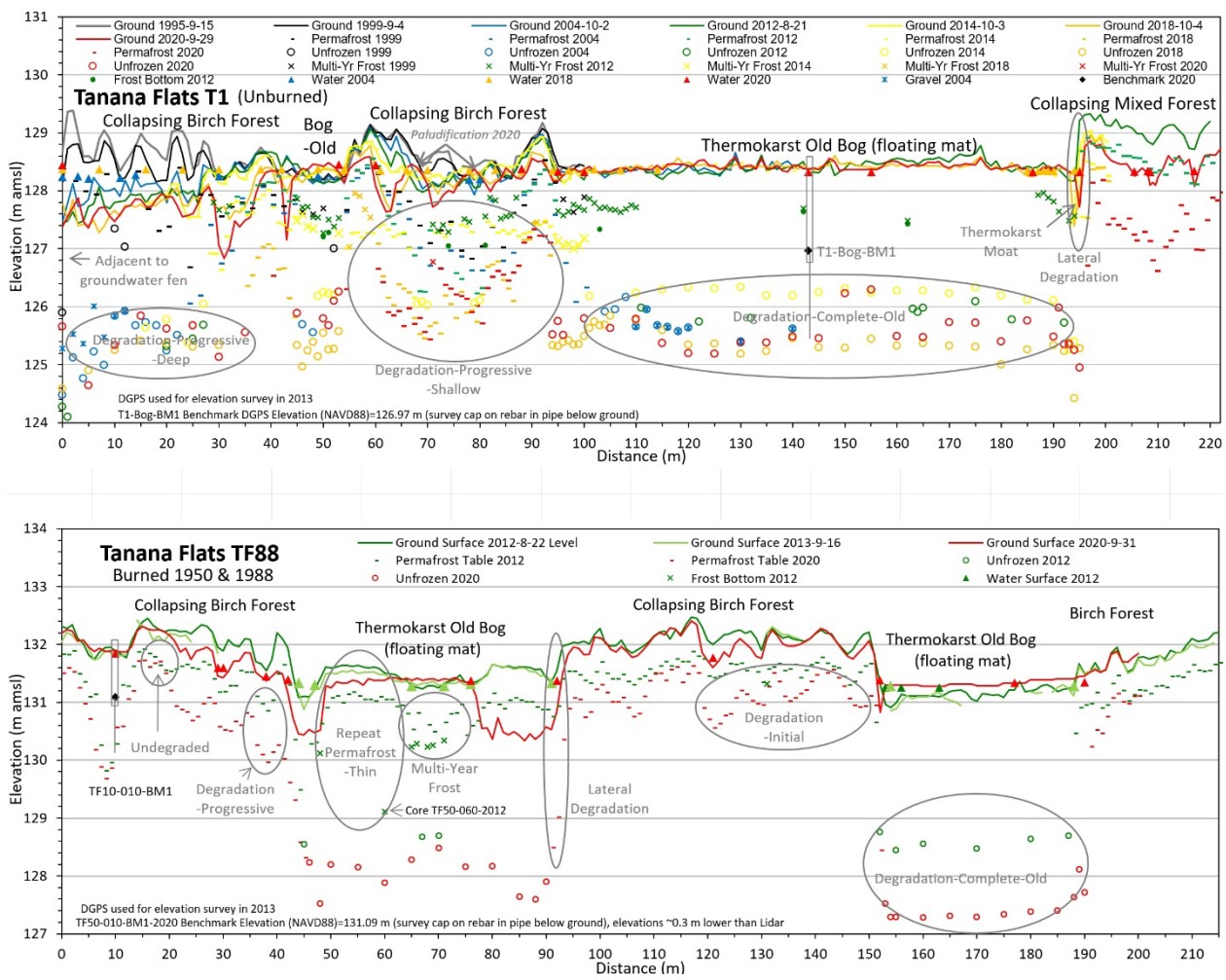

**Figure 2. Topo-profile for T1 (top) and TF88 (bottom) showing elevations of the ground surface, water surface, permafrost table, and maximum-observed unfrozen depths from 1999 to 2020. Also shown are representative transect segments in varying stages of permafrost degradation and aggradation.**

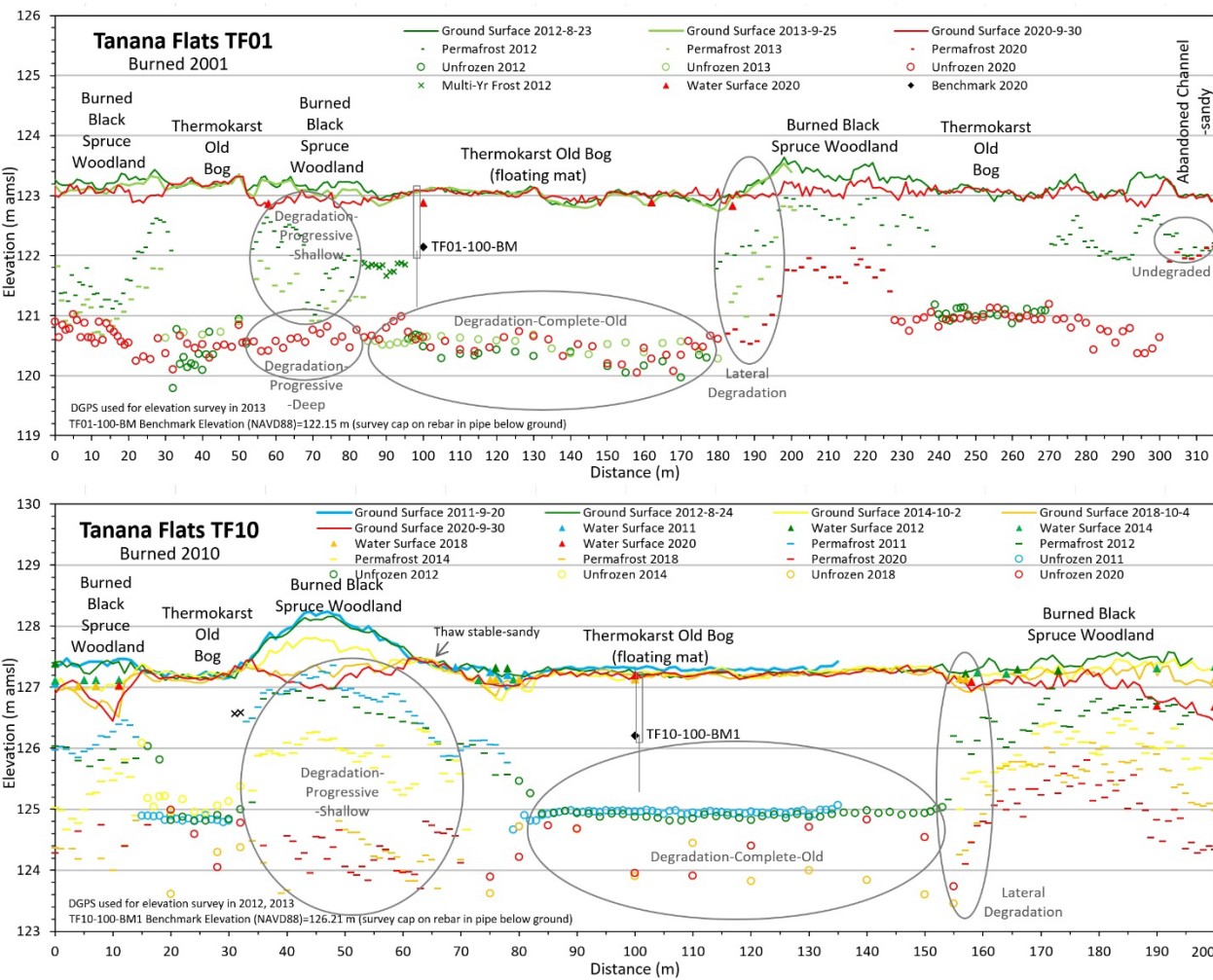

**Figure 3. Topo-profile for TF01 (top) and TF10 (bottom) showing elevations of the ground surface, water surface, permafrost table, and maximum-observed unfrozen depths from 2011 to 2020. Also shown are representative transect segments in varying stages of permafrost degradation and aggradation.**

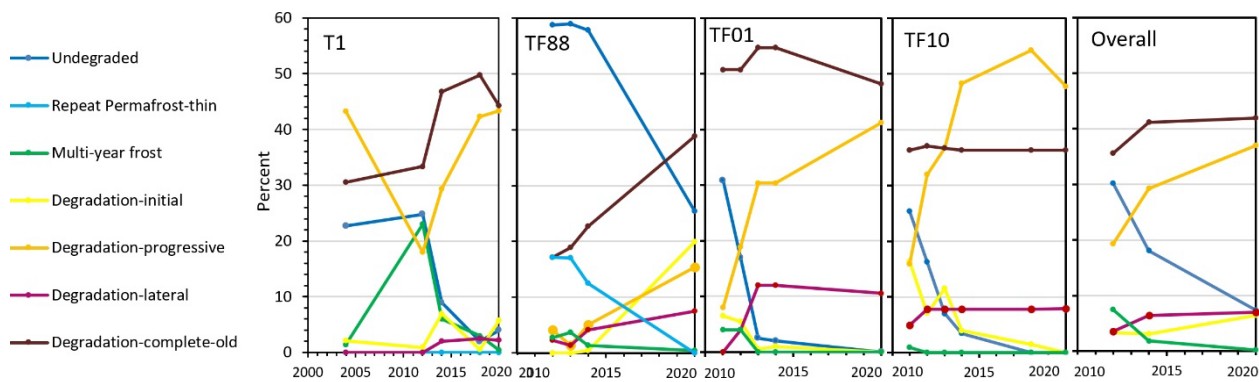

**Figure 4. Trends in extent (percent of transect) of the seven degradation/aggradation stages at four transects (0-200 m) over time.**

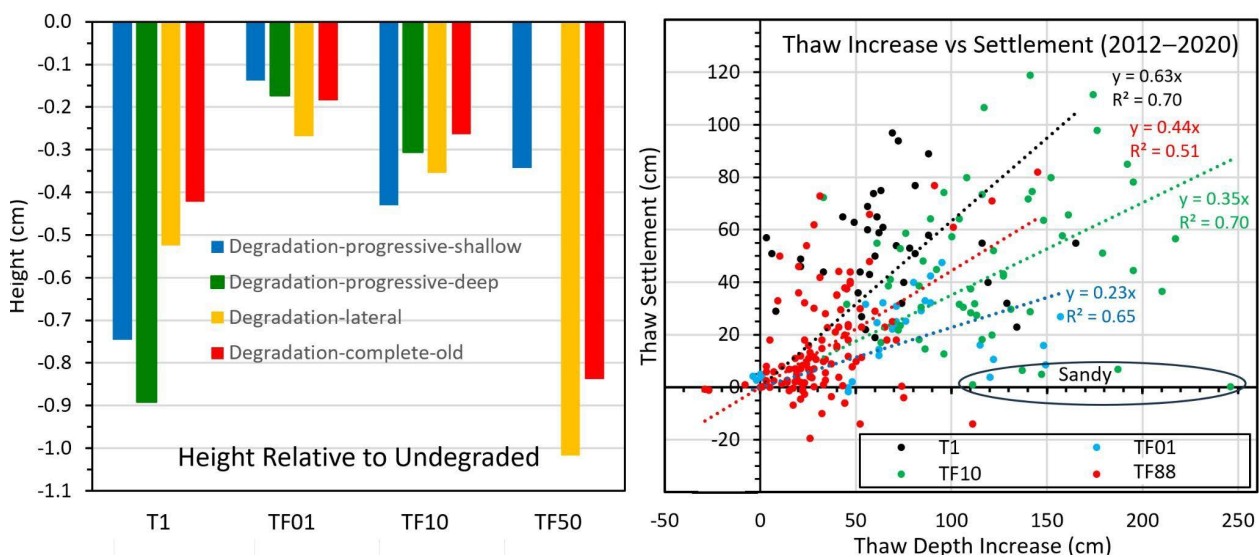

**Figure 5. Change in elevation (cm) of mean surface heights in 2020 by degradation stage relative to mean undegraded height in 2012 (left), and relationship of increased thaw settlement to increased thaw depth from 2012 to 2020 by transect (right). Note that surface elevation of degradation-complete-old includes new peat (floating mat) added after thermokarst and thus does not represent total settlement of the original surface.**

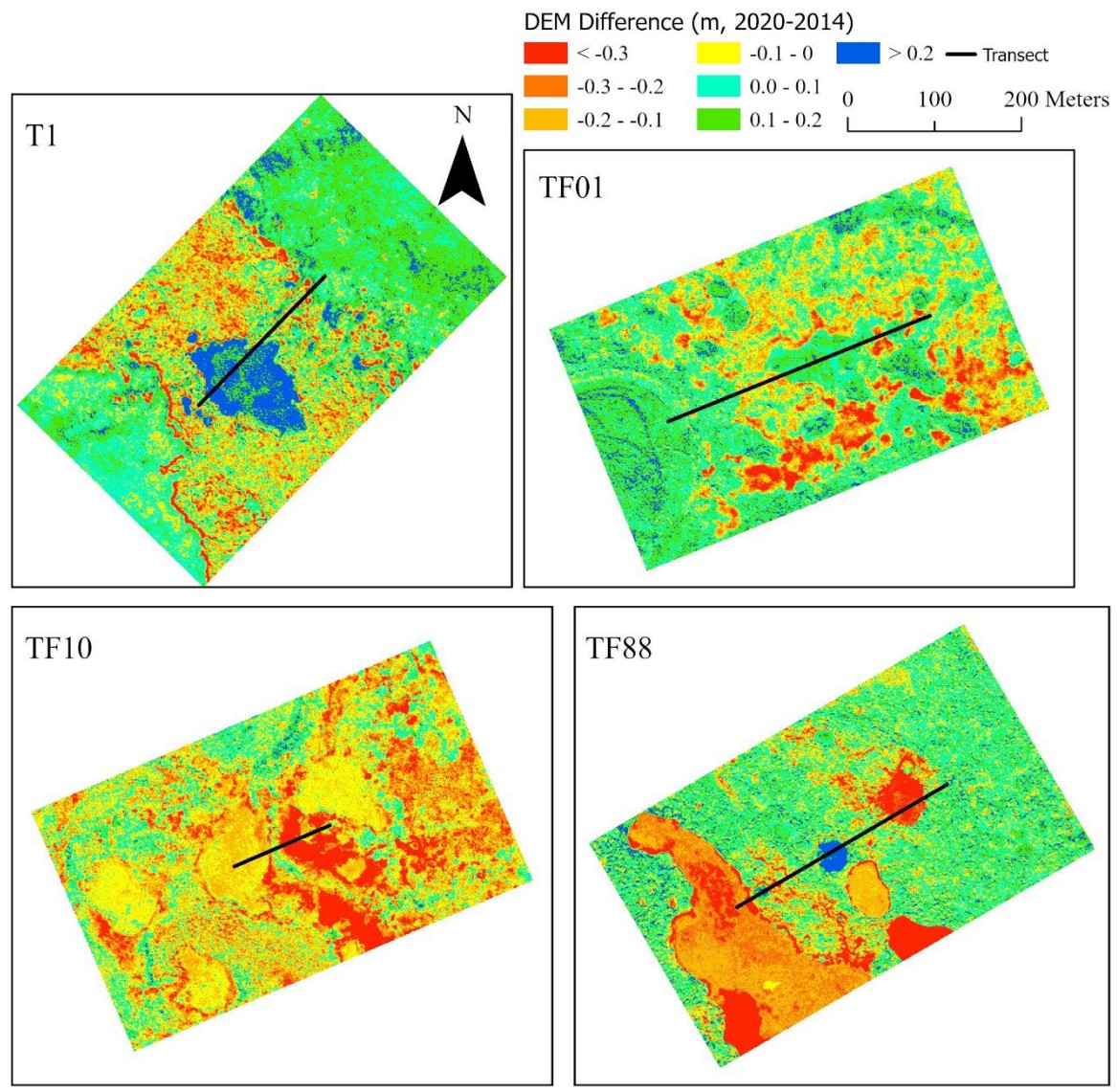

**Figure 6. LiDAR DEM differences between 2014 and 2020 across 300 m by 500 m regions surrounding each of the four transects.**

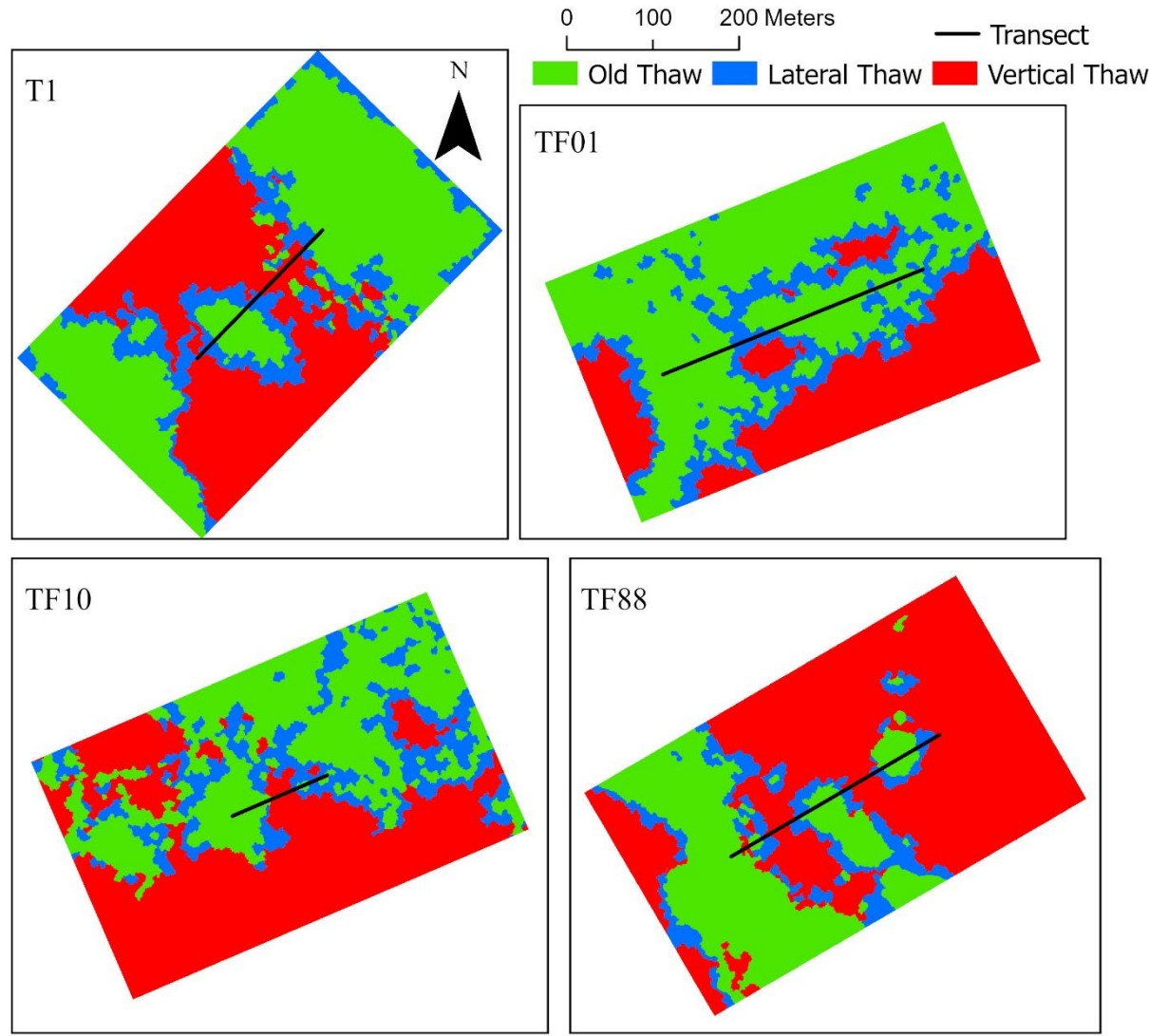

**Figure 7. LiDAR detected thaw degradation stages across 300 m by 500 m regions surrounding each of the four transects between 2014 and 2020.**

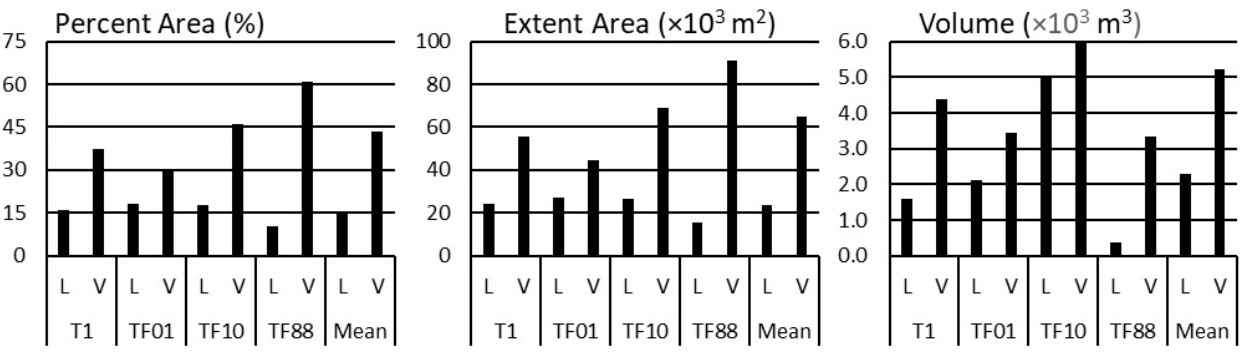

**Figure 8. Thaw extent (percent area and m$^2$) and volumetric settlement loss (m$^3$) due to lateral (L) and vertical (V) thaw derived from repeat LiDAR DEMs for a 300 m by 500 m buffered area around each transect.**

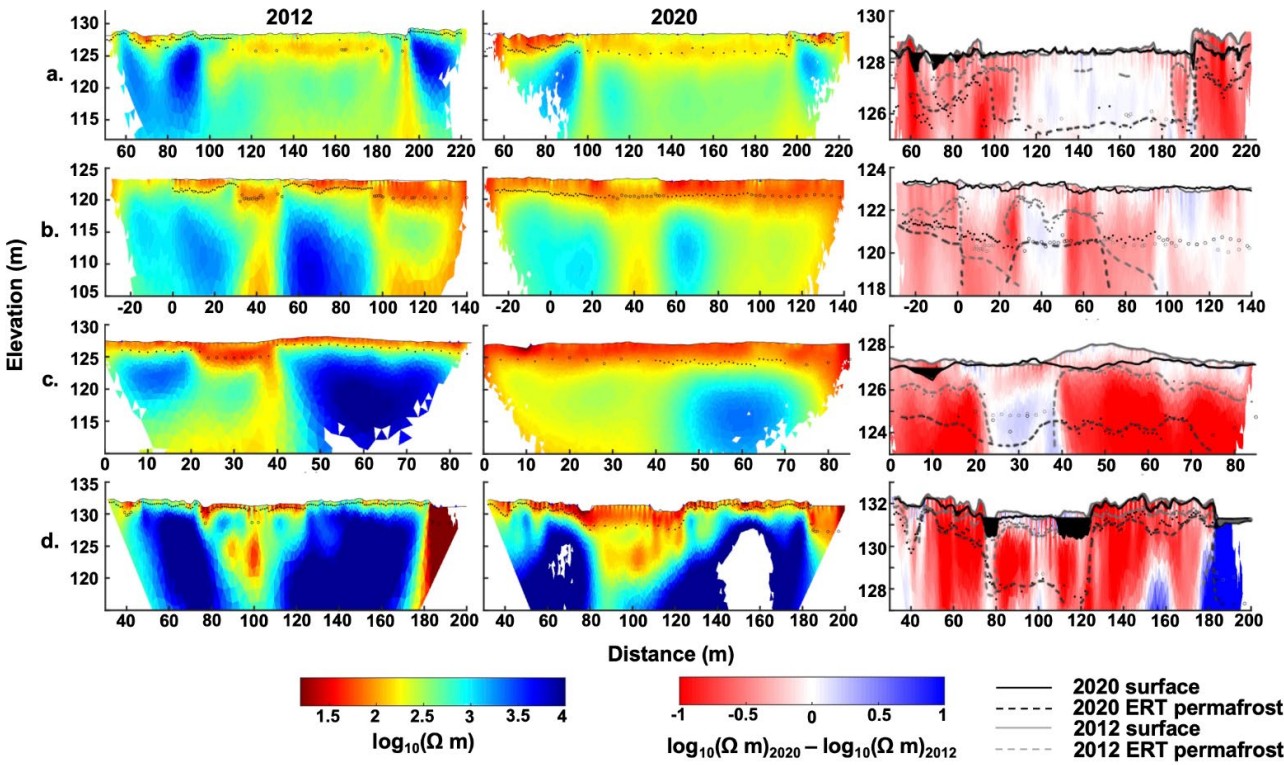

**Figure 9. Repeat ERT data for (a) T1, (b) TF01, (c) TF10, (d) TF88. Left (2012) and middle (2020) show thaw depth (dots), maximum observed unfrozen depth (open circles), and surface water (triangles) atop inverse model results. The right column shows the log difference between 2012 and 2020 plotted on the 2012 inversion mesh. Black and gray lines represent the surface topography and permafrost boundary as interpreted from dipole-dipole ERT surveys and thaw probe measurements.**