# Peer review of "Comparing thaw probing, electrical resistivity tomography, and airborne LiDAR to quantify lateral and vertical thaw in rapidly degrading boreal permafrost"

_EGUsphere, 2024_

## Author Comment (AC1)

Most recent version

Dear Editor-

Thank you for providing these constructive Reviewer comments to our manuscript titled "Comparing thaw probing, electrical resistivity tomography, and airborne lidar to quantify lateral and vertical thaw in rapidly degrading boreal permafrost."

We have provided a tracked change and a clean version of revised manuscripts to Copernicus web site and here we provide detailed information on how we addressed each comment. Original comments are in black, our responses are in blue text, and new/updated contents are provided as red text in quotation marks.

**RC2**: 'Comment on egusphere-2024-3997', Anonymous Referee #2, 17 Apr 2025
GENERAL COMMENTS
The authors use repeat probing, electrical resistivity tomography, and LiDAR surveys to assess lateral and vertical thaw of permafrost across one undisturbed and three fire disturbed transects in the Tanana Flats near Fairbanks, Alaska from 2012 to 2020. The authors use the frost probing results to classify sections of each of the study transects into three permafrost aggradation conditions or five permafrost degradation stages, and they find that the degraded proportion increases through time. The greatest degradation was found to occur at the site that was disturbed by fire most recently. Overall, the manuscript is interesting as it uses three important permafrost monitoring/measurement methods in the assessment of permafrost thaw at different stages following fire disturbance. However, I believe that the manuscript requires major revisions prior to further consideration for publication in The Cryosphere.

- Each of the methods that is used is a recognized method for monitoring or measuring permafrost change through time. However, the authors do not currently provide any information on accuracies, precisions, and/or errors associated with each of these methods. This information needs to be included for the reader to be able to properly assess the significance of thaw/change. For example, the authors use thresholds of 0.3 m for assessing subsidence using LiDAR and of 0.15 m for assessing settlement from ground surface measurements, but there are no descriptions or explanations on the errors of the methods.

Accuracy of the field probing depends on soil materials, unfrozen water content, and depth. As the soils within the top 3 m were mostly peat and silt, shallow (<1 m) probing typically hit a hard refusal boundary (indicating frozen conditions); for these conditions we consider the accuracy to within 3 cm (given a "soft" mossy and litter-rich ground surface). Occasionally in partial degrading permafrost near the surface (within 1.5 m) the probe encountered permafrost with substantial unfrozen water content with refusal gradually becoming harder across an ~20 cm transition zone; we consider the accuracy under these conditions to be with 20 cm. Deep (2 to 5 m) probing is more problematic as friction along the entire probe increases: for these conditions we believe we can reliably detect a frozen boundary within 2 m; by 5 m the resistance is such that probing becomes unreliable. At these depths, we are confident in the determination of unfrozen

conditions, but have only low to moderate confidence is assigning a refusal to be the result of frozen conditions. For these situations we repeated raise and thrust the probe and use our best judgement to assign frozen/unfrozen status. Differentiating persistent seasonal frost from very thin permafrost also is a challenge. Here we are confident that frost <30 cm thick at the end of summer is seasonal frost and frost >50 cm thick is permafrost; for in between thicknesses we have low confidence as to whether it is seasonal or permanent frost. In some areas we noted a thin frost layer (30 to 50 cm) to persist for 2 to 3 seasons, and we refer to this as multi-year frost. This multi-year frost is a common problem for probing ecosystem-driven permafrost in the boreal region.

We conducted detailed LiDAR error analysis in Zhang et al. (2023). The propagated error to LiDAR DEM of Difference (DoD) was 0.13 m based upon the average error of 0.093 m in DEM (95th percentile) across different ecotypes in our study area. The LiDAR error was below the identified threshold of 0.15 m from field measurements. If we consider this error in LiDAR DoD for subsidence analysis, the threshold of 0.3 m was reasonable (0.15 m + 0.13 m, about 0.3 m). We included our LiDAR error results in section 2.2.2 for LiDAR data description. The following text has bene added in section 2.2.2:

"A total of 183 Ground Control Points (GCPs) were measured using real time kinematic and post processed kinematic techniques to validate the accuracy of the LiDAR DEM products. The average error was 0.093 m in DEM (95th percentile) across different ecotypes (Zhang et al., 2023), leading to the propagated error of 0.13 m in the LiDAR DEM of difference. This error was considered in our thaw stage analysis."

- The authors apply annual frost probing, repeat ERT from 2012 and 2020, and repeat LiDAR from 2014 and 2020.
o Frost probing is a standard field method for confirming permafrost presence and for measuring the depth of seasonal thaw on the date of the measurement. The authors are strongly recommended to provide the exact dates of frost probe measurements and to provide contextual information that could support comparisons between measurements from year to year. For example, a sum of the total number of thawing degree days per year up to the measurement date for frost probing, ERT surveys, and LiDAR acquisitions could demonstrate that field measurements were conducted at similar points in the thaw season throughout the study period and that no bias was introduced by the field visit/measurement date.

Good point on providing date information. All frost prober measurements were done in late September of each year as is typical for Interior Alaska (i.e. refer to the Circumpolar Active Layer Monitoring Network (https://www2.gwu.edu/~calm/)). The years surveyed are provided in Figures 2 and 3. However, we do not think an analysis of degree day differences among years is needed or explanatory. Our analysis focuses on the relative extent of the various degradation stages, and increase in progressive, lateral, and complete degradation, which occur over many years. So, the degradation cannot be directly related to thawing degree days for any one year. Note that we do not compare thaw depths for the undegraded stage that might be sensitive to time of probing and thawing degree days.

o   Given that the first ERT survey and the first LiDAR acquisition were conducted in different years, it would be very helpful if the authors could provide a description of the climatic context for the 2012 versus 2014 years. The authors should also include a limitations section in the Discussion that addresses overall limitations to their study, including issues in comparing between methods in different years.

•   The authors have compiled and collected valuable data that describes permafrost conditions under different disturbance contexts in the Tanana Flats. However, much of this work has already been described in previous publications (e.g., Douglas et al., 2016), and the advancement of this contribution relative to previous contributions appears to be the development of a classification scheme that identifies an increase in degradation through time. The classification scheme has eight total classes, three for aggradation (?) and five for degradation. The main message of this paper appears to be focused on trends in the proportions of these classes at each study site through time. This classification scheme is an interesting approach for evaluating broad patterns of permafrost aggradation versus degradation between sites, but it also introduces several issues.

o   First, eight classes is a lot of classes. The authors are encouraged to consider combining some of these classes together and recategorizing them as intact permafrost (UD), newly aggraded permafrost (MF/NPR), vertically degraded permafrost (DI/DPS/DPD), and laterally degraded permafrost (DL). The patterns of decreased intact and newly aggraded permafrost and increased vertically/laterally degraded permafrost at the study sites would still be evident, but having fewer classes would improve readability and overall comprehension.

Given our struggles and more than 25 years of field experiences dealing with the complexity of quantifying boreal permafrost, we prefer to maintain the complexity. But we think the suggestion of combining DPS and DPD is good and we have done that (see updated Figure 4). Distinguishing DI is useful because it is easily reversable by a cold, snow-less winter, whereas, DPS is unlikely to freezeback. We implemented the DPS/DPD combination.

o   Second, the UD class is defined according to a threshold of 0.8 m, which the authors describe as being a typical maximum range of late summer thaw for organic-rich soil. The DI and DPS classes are defined according to a threshold of 1.1 m, and the DPD and DCO are defined according to the limit of the frost probe, which the authors describe as ranging from 2.5 to 4 m. These thresholds are not necessarily applicable to other study regions, nor to other materials within the same study region, and do not take into account settlement of the ground surface. As permafrost thaws, it is possible for ground ice to melt, for the ground surface to settle, and for the overall thaw penetration to increase, but for the frost table to remain similar from year to year. The limit of the frost probe, measuring from 2.5 to 4 m, is also quite a large range, and the authors are encouraged to either be more specific or to use the lower limit of 2.5 m. By combining the DI/DPS/DPD classes together into an overall vertically degraded permafrost class, as recommended above, this would eliminate

the use of the 1.1 and 2.5-4 m thresholds. These changes would make the classification scheme more general and potentially more applicable to other permafrost regions.

We agree with these challenges in identifying, tracking, and categorizing permafrost thaw, particularly the aspects of subsidence. We do account for subsidence in our study (note multiple plots of ground surface elevations by time in Figures 2 and 3). It would be difficult to change the frost probe limit as we had different probes for different years but we want to provide all of our data for all measurements. We disagree with the suggestion to combine the DI/DPS/DPD classes as we feel this identification is indeed applicable to other regions and reflects the Fairbanks sites well. We agree that the different depths we have used to categorize these different degradation classes may not be applicable to other field sites but we would argue strongly that the different classes themselves are quite transferable to other locations. Maybe not all of them at every location but based on our more than 30 years studying these sites we are confident they represent the changes ongoing at these sites well.

- The classification scheme, which makes up the main message of the paper, is derived completely from the frost probing results that occurred annually from 2012 to 2020. However, the title, introduction, and discussion suggest that each of the three methods will be separately evaluated for their utility in monitoring or delineating permafrost. At the very end of the discussion, P22 L497-498, the authors describe the importance of "a coupled application of these methods". The authors are encouraged to follow what they have written here and to actually integrate these methods. Frost probing is an important validation measurement for ERT, as the authors describe in P16 L313-314, and repeat LiDAR acquisitions are very useful for evaluating lateral versus vertical thaw. It would be helpful to present the data in a more integrated way, perhaps by extracting the old thaw/lateral thaw/vertical thaw shown in Figure 7 along the actual frost probing/ERT study transects and comparing those results with the repeat ERT results from Figure 9 and the designation of vertically versus laterally thawed permafrost from the frost probing.

Good point. We provided a detailed classification scheme and included it SI Table 1. The selected each method has its pros and cons, and a combined application of them offers a more effective way to characterize permafrost degradation in multiple dimensions. The idea of applying three approaches is the focus of this work, and the synergy of them offers us the opportunity to detail the thaw stage from field observations to airborne LiDAR mapping and ERT subsurface support. We linked each component of the manuscript such as title, introduction, our data collection, results, and discussion with the idea of this work to give insight of thaw stages and permafrost change across time.

SPECIFIC COMMENTS
TITLE
P1 L1, lidar should be written as LiDAR, as it is actually an acronym for "Light Detection and Ranging".
We have changed this as suggested but note that in other journals/publications both spellings have been used.

"Comparing thaw probing, electrical resistivity tomography, and airborne lidar LiDAR to quantify lateral and vertical thaw in rapidly degrading boreal permafrost"

We have also changed "lidar" and "Lidar" to "LiDAR" throughout the manuscript (~20 locations) for consistency.

ABSTRACT
P1 L10-24, the abstract does not make direct mention of the classification scheme that was developed as part of this study. It does mention the loss of "multi-year frost and repeat thin permafrost", which are part of the classification scheme, but these are not commonly known terms in permafrost science. Please revise to include more information here, explaining the development of the classification scheme.
We are limited to 250 words and the Abstract is currently at 250 words. Two sentences summarize the results for the repeat surveying and classification. Introducing and defining these terms would lead to a far longer abstract and we feel it is more appropriate to provide these definitions in the main text where they are introduced and discussed.

P1 L14, here the repeat measurements of ground surface elevation and depth to the top of permafrost are described as having occurred from 1999 to 2020, but later in the main text, the transects are described as having first been established in 2011 and that surface topography and seasonal thaw depth were measured in fall 2012. I would recommend changing the text in the abstract from 1999 to 2020 to 2012 to 2020 to remain consistent with what is actually presented and described.
We have edited the abstract to read:
"First, repeat measurements of ground-surface elevation and depth to the top of near-surface permafrost were made over an 8-to-21-year period at different sites"

INTRODUCTION
P2 L41, please remove "in the subsurface", as permafrost is already below the ground surface.
This has been changed, as suggested, to:
"These studies rely on differences in surface elevation or changes in vegetation cover over time to identify where permafrost extent has likely changed"

P2 L43, Holloway et al. 2020 does not describe repeat geophysical measurements in lowland sites, but is rather a review paper on post-fire permafrost and ecosystem response.
This reference has been removed and the sentence now reads:
"Linking repeat geophysical measurements with surface and subsurface surveys is valuable for mapping three dimensional changes in permafrost extent at mountain (Mewes et al., 2017; Buckel et al., 2022) and lowland sites (Lewkowicz et al., 2011; Douglas et al., 2016; Minsley et al., 2023)."

P2 L44, consider changing "the greatest means" to "an effective means".
This has been changed, as suggested, to:

"Combining ground-based geophysical measurements with airborne remote sensing observations provides an effective means for quantifying three-dimensional permafrost thaw (Minsley et al., 2015; Uhlemann et al., 2021)."

P2 L52-53, consider changing "in an area of degrading ice-rich lowland permafrost" to "in a lowland area containing ice-rich permafrost".
This has been changed, as suggested, to:
"The focus of this work was to measure three dimensional rates of change in permafrost extent in a lowland area containing ice-rich permafrost in Interior Alaska"

STUDY SITE AND METHODS
P2 L59, consider renaming to "Study area and methods" or "Study sites and methods", as there is more than one study site.
This has been changed, as suggested, to:
"**2 Study sites and methods**"

P3 L64-65, more information is required on the permafrost conditions at the study sites. Given how much prior work has been conducted here at these study sites, it would be helpful if the authors could provide more information on the permafrost conditions, either here in the study area section or in the beginning of the results section. More specific information on permafrost thickness, permafrost temperatures, active layer thicknesses, surficial materials, depth to bedrock, etc. would all be beneficial.
This is a good suggestion., We have edited/expanded the first paragraph in "2.1 Study site climatology, permafrost geomorphology, and fire history" to:
"Our study focused on four transects located along the northern edge of Tanana Flats, a lowland underlain by discontinuous permafrost that stretches from 5 kilometers south of Fairbanks, Alaska to the north slopes of the Alaska Range ~70 km further south (Fig. 1). Tanana Flats is a broad valley covering approximately 6,000 km$^2$ that spreads northward from the Alaska Range to the Tanana River. Geologically, the Tanana Flats is a complex of fluvial deposits associated with a large outwash fan in the western portion of the area and braided floodplain deposits in the northern and eastern portion of the area (Jorgenson et al., 1999; Walters et al., 1998). Much of the area has a typical stratigraphic sequence of peat (0.5-1.5 m), eolian silt (2-3 m), and alluvial sand and gravel (Jorgenson et al., 1999; Brown et al. 2015). Permafrost covers about 44% of the area (Jorgenson et al., 2001). Permafrost is mainly epigenetic and formed during downward freezing. Excess ice contents in birch forests can be greater than 50% while ice contents in the black spruce stands are typically closer to 20% (Brown et al., 2015; Jorgenson et al., 2001; Jorgenson et al. 2025; Walters et al., 1998). Deep boreholes found permafrost at depths ranging from 7.3 m (Ferrick et al., 2008) to 47 m (Jorgenson et al., 2001), while electrical resistivity tomography indicated minimum thicknesses of >20 m were common (Douglas et al., 2015). The mean annual temperatures near the top of permafrost for undegraded permafrost are between 0 and -1 °C (Brown et al. 2015; Jorgenson et al. 2025). Active-layer thickness above permafrost typically ranges from 50 to 75 cm in birch and spruce forests (Brown et al., 2015)."

And added these references:

Jorgenson, M.T., Douglas, T.A., Shur, Y.L., Kanevskiy, M.Z. (2025) Mapping the Vulnerability of Boreal Permafrost in Central Alaska in Relation to Thaw Rate, Ground Ice, and Thermokarst Development. Journal of Geophysical Research: Earth Surface, in press.

Walters JC, Racine CH, Jorgenson MT. Characteristics of permafrost in the Tanana Flats, interior Alaska. InPermafrost: Seventh International Conference, June 1998 (pp. 23-27). Québec, CA: University of Laval.

P3 Figure 1, consider including the location of Fairbanks on the inset map, as the study region is described in the text as being south of Fairbanks, but the location of Fairbanks is not provided.
The location of Fairbanks has been added to the Figure. We have changed the main text to reflect the location more specifically:
"Our study focused on four transects located along the northern edge of Tanana Flats, a lowland underlain by discontinuous permafrost that stretches from 5 kilometers south of Fairbanks, Alaska to the north slopes of the Alaska Range ~70 km further south (Fig. 1)"

P3 Figure 1, consider including some site photos.
We have added site photos to Supplemental Information Figure 3.

P3 Figure 1, please provide the source for the fire perimeters in the figure caption.
We have edited this caption to read:
"Figure 1. Study area in Interior Alaska (a) identifying the regional location in Alaska and (b) the location of four transects: T1 (unburned), TF88 (burned in 1988), TF01 (burned in 2001), and TF10 (burned in 2010). Perimeters for the 1988, 2001, and 2010 fires are also provided from the Alaska Interagency Coordination Center"

P3 Figure 1, please review the accessibility of this figure and the other figures by running them through a colour blindness simulator.
We have done this and modified color schemes accordingly.

P4 L76, is there a particular station within the Tanana Flats at which these values are measured? Please provide the corresponding period for the mean annual air temperature and mean summer and winter monthly air temperatures. For example, do these values correspond to the 1991-2020 climate normal?
The long-term station for the Fairbanks area is Fairbanks International Airport which is 5 km from the northwest boundary of Tanana Flats. We provide specific decades for which the temperature values are indicated. We also provided the climatology of temperature, precipitation and snow depth of our study area in SI Figure 1.

"Typical mean annual precipitation is 28 cm water equivalent with 45% of this as snow (Liston and Hiemstra, 2011). Based on decadal mean annual temperatures at the Fairbanks International Airport the area warmed ~2.3 °C between the 1930s-1940s and 2010-2020. Over that same timeframe mean summer temperatures (May 1 to October

10) warmed ~1.7 °C while mean winter temperatures (October 11-April 30) warmed 2-4 °C (Douglas et al., 2024). ”

Note also the updated to "Douglas et al. in press" to:
"Douglas TA, Barker AJ, Monteath AJ, Froese DG. A local meteoric water line for Interior Alaska constrains Paleoclimate from 40,000 year old relict permafrost. Environmental Research Letters. 2024 Aug 21."
Which has been updated in the reference list.

P4 L76-81, please consider providing additional context related to climatic conditions during the study period. A plot showing mean annual air temperatures and total annual precipitation from 2012 to 2020 would help to contextualize the study.
We cite Jorgenson et al. (2020) repeatedly and this paper provides wonderful summaries of temperature and precipitation in Figure 12 that includes most of the time suggested.
This is a great suggestion. We have added two Figures to the Supplemental Information summarizing the climate conditions that place the ~10 years of the study in perspective. These are called out in the text with this:
"More detailed information on the regional climate, with a focus on Tanana Flats, is provided in Supplemental Information Figs. 1 and 2. The period between 2012 and 2020 (when the majority of the field measurements were collected, including repeat electrical resistivity tomography) included three of the ten warmest/wettest summers and three of the ten warmest wettest winters in the ~100-year record. However, no top ten summers or winters for cool and dry or cold and low snow, favorable to permafrost stability or aggradation, occurred during the same timeframe. Two of the five wettest summers in the entire meteorological record (2014 and 2016) and two of the top three highest mean annual air temperatures (MAATs) were also recorded during the study period. This provided conditions favorable to permafrost thaw."

P4 L83, what is meant by "pure or mixed white spruce"?
We have edited this for clarity to:
"The dominant forest cover includes deciduous Alaska paper birch (Betula neoalaskana) and aspen (Populus tremuloides) mixed with pure or mixed white spruce (Picea glauca) or mixed white and black spruce (Picea Mariana)."

P4 L85, please provide an estimate of the fire interval here, rather than saying "regular intervals".
Thank you for this suggestion. It has been edited to:
"The dominant forest cover on Tanana Flats includes deciduous Alaska paper birch (*Betula neoalaskana*) and aspen (*Populus tremuloides*) mixed with pure white spruce (Picea glauca) or mixed white and black spruce (*Picea Mariana*). Ground cover is dominated by *Sphagnum* spp. In poorly drained areas feather mosses (*Pleurozium schreberi, Hylocomnium splendens*) are common. This land cover is well suited to protect permafrost from warm summers, however, it is also subjected to fire return intervals of 50-130 years with more frequent fires in black spruce stands (Johnstone et

al., 2010; Brown and Johnstone, 2012; Douglas et al., 2014; Brown et al., 2015; Potter and Hugny, 2020). ”

These references have been added:
Johnstone JF, Hollingsworth TN, CHAPIN III FS, Mack MC. Changes in fire regime break the legacy lock on successional trajectories in Alaskan boreal forest. Global change biology. 2010 Apr;16(4):1281-95.

Brown CD, Johnstone JF. Once burned, twice shy: Repeat fires reduce seed availability and alter substrate constraints on Picea mariana regeneration. Forest Ecology and Management. 2012 Feb 15;266:34-41.

P4 L95, is "TF50" supposed to be "TF88"?
Great catch! Thank you. This has been edited to:
"In Douglas et al. (2016) T1 is referred to as "1930" and TF50 is referred to as "1988"."

P4 L87-95, please provide a rationale for the extension of the lengths of TF01, TF10, and T1 from 2011 to 2012.
This has been updated to:
"Transects were positioned to cross a range of permafrost and non-permafrost ecotypes and TF01 and TF10 were extended to more adequately represent different cover types."

P4 L100, there is no mention of the exact timing or dates of the site visits that were completed each year from 2012 to 2020. This information is very important, as biases can be introduced if site visits occurred later in the thaw season as the study period progressed. Please provide a table of the dates of the site visits and additional contextual information, such as the sum of thawing degree days leading up to the site visit date for each year. This can be included as supplementary information if needed.
As was stated earlier we have added the following text:
"Repeated thaw probing using a metal rod in late summer (September) quantified changes in the top of near-surface permafrost. Maximum probing depths varied from 2.5 to 4.0 m depending on the number of extensions used, occurrence of gravel, and stickiness of unfrozen silts."

P4 L99-102, what was used to measure surface topography and elevation? For example, was the elevation estimated from a handheld GPS and the topography then estimated using an Abney level or clinometer? Or was a differential GPS used, as indicated by Figure 2? Please provide a description of the measurement method, along with mention of the accuracy and precision of the measurement method.
We have added tis information here:
"Topographic surveys of ground- and water-surface elevations usually were made using an auto-level and stadia rod at 1-m intervals along the four permanently marked transects. We estimate measurement accuracy to be within 3 cm for firm ground and within 10 cm for soft floating mats in bogs and fens. A survey-grade, differential global position system (DGPS) was used in 2012 at TF10 and in 2013 at TF01, TF10, and TF50 to determine ground elevations using a 15-second observation time at each 1-m

interval. The data were post-processed using data from a base station in Fairbanks. We estimate the accuracy to be mostly with 10 cm in open areas and 30 cm in forested areas. For auto-level surveying, elevations were calculated relative to permanent benchmarks, where the elevations were determined through repeated measurements."

P4-5 L103-118, consider sharing the classification scheme in the form of a table or a schematic.
This is a great suggestion! We have added this Table to the Supplemental Information:

**Supplemental Information**
Table 1.

| Degradation stage | | Code | Description |
|---|---|---|---|
| Undegraded | | UD | Stable permafrost with thaw depths within the range of normal variation. In the mid-boreal region with fine-grained soils, thaw depths typically are <0.8 m |
| Degradation-initial | | DI | The initial stage of permafrost degradation that exceeds the normal variation of stable permafrost, but does not yet create a closed talik, the thin unfrozen zone between the active layer and permafrost table. Thaw depths increased over time to >1.1 m. This stage is more applicable to arctic regions, where permafrost causes shallow depressions. It is particularly useful for situations where the active layer has increased sufficiently to thaw underlying ice wedges. |
| Degradation-progressive | Degradation-progressive-shallow | DPS | The progressive increase in thaw depths below the zone of seasonal freezing and thawing, resulting in the expansion of a closed talik. Depths to the permafrost table can still be measured, but different from DPD where thaw probing no longer encounters the permafrost table. This is useful because increases in thaw depths for DPS can still be calculated. For warm permafrost in mid-boreal regions, thaw depths >1.1 m typically indicate a closed talik. Thaw probing to depths of 3 m in fine-grained soils typically is reliable. |
| | Degradation-progressive-deep | DPD | Similar to above, except thawing probing does not encounter the top of permafrost. This stage typically is limited to small surface depressions <10 m across. Large features may be Degradation-Complete |

| | | |
|---|---|---|
| Degradation-lateral | DL | Lateral degradation is used to define permafrost thaw that occurs along margins of permafrost plateaus and thermokarst features. The lateral thaw can cause a sloping boundary near the top of permafrost or a thermal niche well below the permafrost table due to warm subsurface water temperatures. This stage can occur even under cold climates. |
| Degradation-complete | DCO | This stage is associated with open taliks where permafrost has thawed completely through the permafrost zone. It can sometimes be detected with electrical resistivity tomography when permafrost is relatively thin (<30 m thick). More typically, DC is assumed to be present under large thermokarst bogs and fens. |

And the text introducing the classification scheme has been edited to:
"We differentiated three quasi-stable permafrost conditions and four degradation stages using a system modified from Jorgenson (2021) to address the complicated nature of permafrost formation and degradation in boreal ecosystem-driven permafrost (Supplemental Information Table 1; Supplemental Information Fig. 5)"

P4-5 L103-118, the three "permafrost conditions" are later described in the caption for Figure 4 as "aggradation stages" and are earlier described in the abstract as "types of permafrost aggradation", but these are not really aggradation processes. Please revise.
Figure 4 has been updated by combining the "Degradation progressive deep" and "degradation progressive shallow" into one stage called "Degradation progressive."

P4 L106-107, please provide a reference for the threshold of 0.8 m as the typical maximum range of late summer thaw for organic-rich soil.
We have edited this to:
"For permafrost types, undegraded (UD) was assigned when thaw depths were less than 0.8 m (typical maximum range of late summer thaw for organic-rich soil in the Fairbanks area; Douglas et al., 2021)."

And added this reference:
Douglas TA, Turetsky MR, Koven CD. Increased rainfall stimulates permafrost thaw across a variety of Interior Alaskan boreal ecosystems. NPJ Climate and Atmospheric Science. 2020 Jul 24;3(1):28.

P5 L131-132, please provide a reference for this statement on the dipole-dipole array.
We have added references as follows:

"A dipole-dipole measurement geometry was used for all ERT surveys due to its sensitivity to lateral features and our focus on changes in near-surface permafrost distribution (Douglas et al. 2016; Minsley et al., 2022). "

P6 L166, please provide a brief description and a reference for this statement on salt and pepper effects.
We have edited the sentence to address this:
"This object-based analysis approach reduces "salt-and-pepper" effects which are caused by isolated pixels with high spatial heterogeneity. This abnormality is considered as noise affecting analysis accuracy and results (Blaschke et al. 2000)."

And the reference:
Blaschke, T., S. Lang, E. Lorup, J. Strobl, and P. Zeil. 2000. Object-Oriented Image Processing in an Integrated GIS/Remote Sensing Environment and Perspectives for Environmental Applications. Environmental Information for Planning, Politics and the Public 2: 555–570.

RESULTS
P7 L173, here there are six degradation stages that are mentioned, but the earlier classification scheme only describes five stages? Please revise and correct.
As per another Reviewer suggestion there are now a total of seven degradation/aggradation stages and a total of four degradation stages (Figure 4).

P7 L173-187, this section repeats the descriptions for the different classifications, which were first described on P5. The descriptions for the classifications here on P7 are also not consistent with those on P5. For example, here, DI is described as regions where thaw depths are >100 cm and < 120 cm or had increased by 30 cm over time. On P5, DI is described as regions where thaw depths increase to 1.1 m. Please ensure that the classes are consistently described and applied, and please revise to remove repetition between sections.
As per this Reviewer's earlier suggestion there are now a total of six degradation/aggradation stages (Figure 4). The text where they are summarized in this paragraph has been summarized/shortened considerably. We agree that reducing repetition is important, however, we feel that it is worth providing a short reminder of each stage before it is summarized.
"We compiled the repeat surveys of surface topography, water depths, thaw depths, and ancillary subsurface measurements from the four transects to define trends of subsurface thaw and corresponding ground surface settlement between 2012 and 2020 (Figs. 2 and 3). We differentiated trends across each transect into seven degradation/aggradation stages (Supplemental Information Table 1). UD (thaw depths remained <100 cm and annual changes <30 cm) occurred along 29% of the transects in 2012 and decreased to 8% by 2020 (Figs. 2 and 3). DI (thaw depths >100 cm and <120 cm or had increased by >30 cm as a brief transitional stage) occurred along 36% of the transects. DPS (vertically increasing thaw depths >120 cm but not more than ~250-300 cm), increased from 12% to 20% between 2012 and 2020. DPD (thaw depths increased to >250-300 cm) increased from 6 to 16%. Together, these shallow and deep

progressive degradation stages (i.e., top-down thaw of near-surface permafrost) increased from 18% to 36% over the 8 year study period. We presume DPD denotes areas with open taliks. DL (lateral thaw bogs and fen margins), increased slightly from 3% to 6%. Regions characterized as DCO occurred under old bogs and fens and presumably had completed degradation through the entire permafrost zone to form through taliks. They increased slightly between 2012 and 2020 from 35% to 44%. We attribute this change, however, primarily due to the loss of Repeat-Permafrost-Thin (RPT) and MF (Figs. 2 and 3) near the surface. MF was used to differentiate areas with a thin layer (typically <30 cm) of frozen ground that persisted for 14 years (permafrost is conventionally defined as frost persisting more than 2 years). MF decreased from 7% to 0%."

P8-9 Figures 2-3, the figures are very comprehensive; however, they are also a bit confusing as they provide so much information. It would be best to perhaps show only the relevant information in 2012 and 2020, as annual information is provided in Figure 4. Please also consider combining Figures 3 and 4 into one large figure with all four sites together and re-organizing the figures to present burned site results in chronological order of fire disturbance: TF88, TF01, TF10. This should help with interpretation of inter-site comparisons/results.
We respectfully disagree with this suggestion. Agreed that there is a lot in here but we feel that these Figures encapsulate the major summary information from discreet surveys taken over time to show change. There are other Figures that follow that provide more succinct summaries on most of the measurements provided in these Figures. We also feel that this presentation is the best way to place the changes over time into one easily viewed image for each transect.

P13 L263-267, what is the accuracy of the elevation measurement method? Are these elevation difference thresholds of 0.15 m beyond the error of the instrument/method? Please clarify.
As per this Reviewer's earlier comment we have added the following paragraph on accuracy into 2.2.1 Field sampling design and measurements:
"Topographic surveys of ground- and water-surface elevations usually were made using an auto-level and stadia rod at 1-m intervals along the four permanently marked transects. We estimate measurement accuracy to be within 3 cm for firm ground and within 10 cm for soft floating mats in bogs and fens. A survey-grade, differential global position system (DGPS) was used in 2012 at TF10 and in 2013 at TF01, TF10, and TF50 to determine ground elevations using a 15-second observation time at each 1-m interval. The data were post-processed using data from a base station in Fairbanks. We estimate the accuracy to be mostly with 10 cm in open areas and 30 cm in forested areas. For auto-level surveying, elevations were calculated relative to permanent benchmarks, where the elevations were determined through repeated measurements."

P14 Figure 7, please show the location of each of the transects within their respective 300 by 500 m LiDAR study areas.
We have provided an updated Figure 7 with the locations of the transects added. They are also added to Figure 6.

P15 L290, please provide additional information on the overall thickness of permafrost along each of the study transects. The authors do state in the study area section that permafrost in this overall region can measure up to 50 m in thickness, though it would be helpful to know how thick the permafrost is at each study site. For example, do the ERT surveys show permafrost thicknesses extending beyond 20 m?

Thank you for this comment. We acknowledge that permafrost depth is an important parameter in permafrost ecosystems, and there are is only one borehole from this area suggesting permafrost depths up to 50 m (Chacho, Arcone, & Delaney, 1995). Additionally, there are no published remotely sensed datasets sensitive to the lower boundaries of permafrost in this region. ERT surveys can yield depth of investigations (DOI) up to 20% of the transect length under ideal conditions (Binley & Kemna, 2005). Our transect lengths for T1, TF01, and TF88 were 168 m with associated maximum DOIs of roughly 34 m, and the length of transect TF10 was 84 m long corresponding to a DOI of around 17 m. Given the low-resistivity layer (the thawed active layer) in the near surface further limiting inverse model sensitivity at depth and knowledge that Dipole-Dipole measurement geometries are most sensitive in the near-surface (Oldenburg & Li, 1999), we are reluctant to interpret the bottom of permafrost from these inverse model solutions.

To address the importance of the lower-permafrost boundary, we have added the following text to the manuscript:

"Various ERT measurement sequences—i.e., permutations of current-injecting and potential electrode pairs—present tradeoffs in lateral versus vertical sensitivity and data acquisition efficiency (Binley and Kemna, 2005). The dipole–dipole configuration employed in this study is particularly sensitive to lateral resistivity contrasts and near-surface features, offering the advantage of relatively rapid acquisition times (Oldenburg and Li, 1999) and validation with thaw probe measurements. While model solutions derived from our ERT data suggest permafrost at depths exceeding 20 m (Fig. 9), the validity of such features remains uncertain in the absence of borehole corroboration. Given the known limitations of dipole–dipole geometries at depth, compounded by conductive near-surface layers that attenuate sensitivity with depth, we refrain from interpreting the lower boundary of permafrost in this dataset. Instead, we emphasize that strong confidence exists in detecting top-down thaw and near-surface changes, which align with ground-based probing across all transects."

And added the following references:
Oldenburg, D.W., Li, Y. (1999), Estimating depth of investigation in dc resistivity and IP surveys, GEOPHYSICS 64: 403-416. https://doi.org/10.1190/1.1444545

Binley, A., Kemna, A. (2005). DC Resistivity and Induced Polarization Methods. In: Rubin, Y., Hubbard, S.S. (eds) Hydrogeophysics. Water Science and Technology Library, vol 50. Springer, Dordrecht. https://doi.org/10.1007/1-4020-3102-5_5

P15 L299-300, this statement that the resistivity values decreased between 2012 and 2014 should be removed if the authors do not present the tomograms from 2014 in this paper.

The tomograms for both 2012 and 2020 are presented in Figure 9 along with the difference. The difference plots show that decreases in resistivity values are dominant across all four repeated transects.

P16 L313-314, great, frost probing is an important measurement to collect to support interpretations of permafrost presence from ERT!

We wholeheartedly agree!

DISCUSSION
P17 L325, change "Interior Alaska permafrost" to "Permafrost in interior Alaska".

This has been changed, as suggested, to:

"Permafrost in Interior Alaska has been slowly thawing for the past ~500 years with sporadic periods of accelerated thaw typically attributed to wildfire and subsequent permafrost stabilization or aggradation associated with forest succession (Jorgenson et al. 2001; Jones et al., 2013)."

P17 L333, this part of the discussion mentions that the study sites consist of mixed forest, birch forest, or black spruce woodland; however, the study area section does not mention black spruce and rather describes the region as containing deciduous Alaska paper birch and aspen mixed with pure or mixed white spruce. Please revise and correct.

We have edited this sentence to:

"All four transects contain low lying old thermokarst bogs surrounded by permafrost plateaus consisting of birch or aspen forests, black or white spruce woodland, or mixed deciduous and conifer forest (Figs. 2 and 3)."

To address a previous comment, we clarified the presence of white and black spruce in Section 2.1 Study site climatology, permafrost geomorphology, and fire history.

P17 L333, "These are the most common landforms above boreal discontinuous permafrost" – what does this mean? That the permafrost plateaus mentioned in the previous sentence are the most common landforms? Please clarify.

We have clarified this sentence to:

"The plateaus, the most common landforms above boreal discontinuous permafrost, are a signature indicator of the presence of permafrost in the region."

P17 L337, what is "braided ice"? I am not familiar with this term as a cryostructure, and it does not seem to be a term that is commonly used outside of this author team (see Brown et al., 2015; Douglas et al., 2016). Please clarify or describe what braided ice is.

We have clarified this as follows:

"This is likely due to lower ice content lenticular ice from 1.2 to 2.8 m depth at TF88 compared to braided (reticulate-platy) ice from 1.5 to 2 m at T1 (Brown et al., 2015)."

P17 L352-354, many years of personal experience with frost probing in various materials would lead me to disagree with these statements. I would say that frost probing does not provide precise measurements, as the depth to the frost table may vary according to the person performing the probing, to the material, and to the timing of the measurement from year to year. I also do not think that it is possible to detect precise changes in soil texture with depth, and if this is the method that is used to determine stratigraphy, then soil pits need to be dug to verify these interpretations. The discussions of the limitations of probing on P18 L370-373 also contradict this description of probing as a "precise" method that can "detect marked changes in soil texture".

This is a great point and we have addressed it by clarifying WHERE the frost probe measurements are most useful and HOW they have been confirmed with soil pits and cores:

"Thaw probing, accompanied by surveying of ground and water-surface elevations, yielded the most precise measurements of subsurface changes in the permafrost table than ERT or repeat LiDAR and this helped identify degradation stages (Figs. 2 and 3). Probing can detect changes in soil texture (peat, silt, sand, gravel) with depth, however, soil stratigraphy was confirmed with soil pits and boreholes (Brown et al., 2015)."

P19 L391, how was the threshold of 0.3 m for significant permafrost thaw determined? What is the error of the LiDAR method? Please include this information.

We used 0.15 m as the threshold for degradation progress identification based upon field measurements, and elevation change larger than it means significant permafrost degradation, and thus using 0.3 m is reasonable when LiDAR DEM of difference was used for identifying such degradation. See above our response to this same comment and here is the new text we have added as clarification:

In 2.2.2 Airborne LiDAR data acquisition and analysis we have added:

"A total of 183 Ground Control Points (GCPs) were measured using real time kinematic and post processed kinematic techniques to validate the accuracy of the LiDAR DEM products. The average error was 0.093 m in DEM (95th percentile) across different ecotypes (Zhang et al., 2023), leading to the propagated error of 0.13 m in the LiDAR DEM of difference. This error was considered in our thaw stage analysis."

P19 L395, these are important limitations to the LiDAR method. Please consider providing context for precipitation in the two years of acquisition (2014, 2020).

This is a relevant comment. To address it we have added some text to the end of this paragraph:

"If the original ground surface settles below the water level or precipitation increases markedly from one acquisition to the other, thaw settlement can be masked by surface water. For example, the summer of 2014 had more total wet precipitation than the summer of 2020 but far more rain fell from late July through September in 2020 than in 2014. Hence the water table elevations of bogs were higher in 2020 than in 2014 (Fig. 6). This elevated water table masks the water feature boundaries that were present in 2014."

P19 L399-403, these sentences are quite repetitive and a bit contradictory, as L399 states that repeat ERT can delineate permafrost boundaries, but L403 states that ERT is less precise in delineating boundaries.

Thank you for pointing this out. The main discrepancy is that we hope to show ERT does well at identifying horizontal changes but does not do as well with lateral changes. To address this, we have edited the paragraph to:

"Repeat ERT (Fig. 9) has a strong advantage in delineating the upper boundary of near-surface permafrost down to depths of 20 m or more. As such, this method is particularly useful at identifying top-down degradation and zones of deep progressive thaw, however, vertical changes in permafrost extent or stratigraphy should be confirmed with boreholes where possible. In interior Alaska resistivity values above 600-800 Ωm (log$_{10}$ resistivity values of 2.8 to 2.9 Ωm) correlate with conductive permafrost material (Hoekstra and McNeill, 1973; Douglas et al., 2008; 2016; Minsley et al., 2022). At our sites the top of near-surface permafrost corresponds with sudden decreases in log$_{10}$ resistivity from values greater than 2.5 Ωm above the permafrost (active layer) to values between 3 and 4 Ωm where permafrost is present. Old thermokarst bogs are readily identified by low resistivity values and permafrost plateaus have rapid vertical changes in resistivity over the upper ~3 m that are confirmed as permafrost by thaw probing. Resistivity values decreased between 2012 and 2020 for every measurement location across all transects at depths of up to ~20 m. The difference map between the two resistivity campaigns shows the greatest decreases in resistivity occurred in permafrost plateaus and this supports surface surveys along the transects (Figs. 2 and 3)."

We deleted this sentence to reduce repetition:
"Top-down thaw of permafrost plateaus and lateral thaw along bog margins is evident."

P19 L404, replace "it's" with "its".
This has been changed to:
"However, ERT is less precise in delineating lateral changes in permafrost bodies, presumably in part due to its sensitivity to unfrozen-water content."

P19 L403-405, these sentences are quite repetitive as well, mentioning the importance of boreholes in both sentences. Please revise.
These have been removed and mention of boreholes has bene provided earlier in this paragraph but only once.

P20 L442-447, this information would actually fit best in the study area section, as it provides helpful context on recent climate history for this region. Please consider moving this.
We respectfully disagree with this suggestion as we feel that here, towards the end of the Discussion, it is best to frame the results showing permafrost degradation at the sites into this broader climate framework. If the Reviewer or Editor are adamant about moving these few sentences into the study area section. We note that we do present a summary of data supporting climate warming in the area in the Study site and Methods Section.

P21 L458-460, this information on sediments at the study sites would also fit well in the study area section. Please consider moving this.

This is a great suggestion and we have moved these two sentences to the end of the last paragraph of **2.1 Study site climatology, permafrost geomorphology, and fire history**

However, we did edit/clarify this information by editing a sentence that was already in **4.4 Factors Affecting Degradation and Aggradation:**

"The sandier soils and lower ice contents at TF01 and TF10 contributed to more rapid thaw and lower thaw settlement than at the more ice rich and peaty transects T1 and TF88."

P21 L476-477, please provide a reference for this statement that describes a positive feedback that facilitates further lateral thaw.

This sentence has been changed to:

"Initially, during collapse of permafrost terrain in flat lowlands, water impounds at the surface providing a positive feedback facilitating further thaw-induced lateral expansion (Westermann et al., 2016; O'Neil et al., 2023)."

And these references have been added:

Westermann S, Langer M, Boike J, Heikenfeld M, Peter M, Etzelmüller B, Krinner G. Simulating the thermal regime and thaw processes of ice-rich permafrost ground with the land-surface model CryoGrid 3. Geoscientific Model Development. 2016 Feb 8;9(2):523-46.

O'Neill HB, Smith SL, Burn CR, Duchesne C, Zhang Y. Widespread permafrost degradation and thaw subsidence in northwest Canada. Journal of Geophysical Research: Earth Surface. 2023 Aug;128(8):e2023JF007262.

CONCLUSION

P22 L494, what data was provided from 2004? The methods section describes the survey transects as being established in 2011/2012, not 2004? Please revise and correct.

We have updated the Methods to clarify when each transect was initially surveyed or modified (expanded):

"The Tanana Flats lowland has experienced numerous wildfires and we established transects to represent high severity fires in the summers of 1988 (TF88, 64.734 °N, 147.826 °W), 2001 (TF01, 64.644 °N, 148.295 °W), and 2010 (TF10, 64.716 °N, 148.010 °W). Transects were initially studied to assess the effects of fire on permafrost (Nossov et al., 2013): TF88 (200 m; first surveyed in 2012) is in an area burned in ~1950 that reburned in 1988; TF01 (established as a 320 m transect in 2012) is in an area burned in 2001, and TF10 (initially 100 m in 2011, extended to 200 m in 2012) is in an area burned in 2010. Transects were positioned to cross a range of permafrost and non-permafrost ecotypes and TF01 and TF10 were extended to more adequately represent different cover types. We also included an unburned site (T1, 64.722 °N, 147.959 °W) that has not burned in recent years (~1950s-present) for comparison.

Supplemental Information Figs. 3 and 4 provide photographs of the field sites. T1 (initially 100 m in 1995, extended to 200 m in 2012) was established during ecological land surveys (Jorgenson et al. 1999). In Douglas et al. (2016) T1 is referred to as "1930". Two of the sites (T1 and TF88) were relatively ice-rich with thick peat and silts extending down 34 m whereas the other two study sites (TF01 and TF10) had sand and gravel at relatively shallow depths and lower ice contents."

To address the specific sentence in question, we have edited it to:
"For example, at site T1, our control transect, top-down thaw of near surface permafrost occurred over 44% of our study sites between 2004 and 2020."

P22 L502, which one field site is this referring to? Please provide the name of the site that is experiencing substantial vertical thaw.
This is at TF88. The sentence has been clarified to:
"Repeat airborne LiDAR also identified lateral and vertical loss of near-surface permafrost at all our sites with 60% of the area at TF88 site exhibiting vertical thaw (Fig. 8)"

P22 L502-504, given the timing of the field studies relative to the timing of the fire disturbances, it is a bit difficult to really determine whether the thaw that has occurred at TF88, TF01, and T1 is associated with the press disturbance of climate warming rather than their respective pulse disturbances. Please consider rewording.
Thank you for this comment. We have edited mention of this to:
"Much of the permafrost thaw at our sites is associated with the press disturbance of climate warming (Douglas et al., 2021; Farquharson et al., 2022). However, rapid loss of near-surface permafrost was initiated immediately after the pulse disturbance of the 2010 fire (TF10; Fig. 3) with subsidence of up to 1 m in the subsequent decade."

P22 L510-512, agreed that it is very challenging to monitor permafrost! But integrating these three methods is a great way to start.
We thank the Reviewer for this and all their other comments.

---

## Author Comment (AC2)

Most recent version

Dear Editor-
Thank you for providing these constructive Reviewer comments to our manuscript titled "Comparing thaw probing, electrical resistivity tomography, and airborne lidar to quantify lateral and vertical thaw in rapidly degrading boreal permafrost."

We have provided a tracked change and a clean version of revised manuscripts to Copernicus web site and here we provide detailed information on how we addressed each comment. Original comments are in black, our responses are in blue text, and new/updated contents are provided as red text in quotation marks.

**Di Wang**
RC1: 'Comment on egusphere-2024-3997', Di Wang, 31 Mar 2025
This is interesting and timely work. Some comments:
1. The description of permafrost degradation is quite general, please provide some data (temperature increasing rate, extreme climate event frequency and severity) to support it.
We are not clear where the Reviewer suggests adding this information but we have edited the first paragraph of the Introduction and provided a new figure in SI (Figure 1) of climatology of air temperature, precipitation and snow since 1900 for our study area to support this:
"Permafrost is warming and degrading across earth's high latitudes. In interior Alaska this includes top-down thaw of near-surface permafrost (Douglas et al., 2021), increased permafrost temperatures and the formation of unthawed zones (taliks; Farquharson et al., 2022), and lateral expansion of thawed areas (Jorgenson et al., 2020). Mean annual air temperatures in the region have increased ~3C since the 1970s (Douglas et al., 2025) and summers are getting wetter (Jorgenson et al., 2020). The ongoing permafrost degradation affects hydrology (Marshall et al., 2021), ecological processes (Foster et al., 2019; Mekonnen et al., 2019), the carbon cycle (Douglas et al., 2014) and infrastructure (Hjort et al., 2022). With warming projected to accelerate over coming decades the spatial extent of permafrost thaw is expected to increase (Wolken et al., 2011)."

2. The literature review is more like a background description. Please explicitly highlight the novelty and necessity of combining thaw probing, ERT, and airborne lidar.
To address this, we have added the following to the end of the third paragraph of the Introduction. We also have detailed discussion of the pros and cons of each method and the effectiveness of a combined application of these three methods in the Discussion section 4.1:
"Combining ground-based geophysical measurements with airborne remote sensing observations provides an effective means for quantifying three-dimensional permafrost thaw (Minsley et al., 2015; Uhlemann et al., 2021). These typically include site-based

surface measurements of the depth of maximum summer season thaw (active layer) and cores of permafrost to quantify ice content."

3. It is not clear why these three technologies were selected. Please revise it.
We have added this sentence to the last paragraph in the Introduction:
"We focus on thaw probing, ERT, and airborne LiDAR methods because they are among the widest applied techniques to measure permafrost degradation."

4. The data collection periods are different among these three technologies. How do you compare them properly?
As we state in this revised version of the manuscript, we do not specifically apply the three methods to the same measurements/applications as they have their unique pros and cons. We have clarified some of the survey information to address this comment and some others. Specifically, in Section 2.2. Data collection, processing, and analysis. In 2.2.1 Field sampling design and measurements we have added this paragraph about topographic surveys:
"Topographic surveys of ground- and water-surface elevations usually were made using an auto-level and stadia rod at 1-m intervals along the four permanently marked transects. We estimate measurement accuracy to be within 3 cm for firm ground and within 10 cm for soft floating mats in bogs and fens. A survey-grade, differential global position system (DGPS) was used in 2012 at TF10 and in 2013 at TF01, TF10, and TF50 to determine ground elevations using a 15-second observation time at each 1-m interval. The data were post-processed using data from a base station in Fairbanks. We estimate the accuracy to be mostly with 10 cm in open areas and 30 cm in forested areas. For auto-level surveying, elevations were calculated relative to permanent benchmarks, where the elevations were determined through repeated measurements."

And, further down:
"Accuracy of the field probing depends on soil materials, unfrozen-water content in partially thawed soils, and depth. As the soils within the top 3 m were mostly peat and silt, shallow (<1 m) probing typically hit a hard refusal boundary (indicating frozen conditions) for stable permafrost; for these conditions we consider the accuracy to within 3 cm (given a "soft" mossy and litter-rich ground surface). Occasionally, in partially degrading permafrost near the surface (within 1.5 m) the probe encountered frozen ground with substantial unfrozen water content with refusal gradually becoming harder across a ~20 cm transition zone; we consider the accuracy under these conditions to be with 20 cm. Deep (2 to 5 m) probing is more problematic as friction along the entire probe increases: for these conditions we believe we can reliably detect a frozen boundary within 2 m, but by 5 m the friction/stickiness is such that probing becomes unreliable. At these depths, we are confident in the determination of unfrozen conditions, but have only low to moderate confidence assigning a refusal to be the result of frozen conditions. For these situations we repeatedly raise and thrust the probe downward and used our best judgement to assign frozen/unfrozen status. Differentiating persistent seasonal frost from very thin permafrost also is a challenge. Here we are confident that frost <30 cm thick at the end of summer is seasonal frost and frost >50 cm thick is permafrost; for in between thicknesses we have low confidence as to

whether it is seasonal or permanent frost. In some areas we noted a thin frost layer (30 to 50 cm) to persist for 2 to 3 seasons, and we refer to this as multi-year frost. This multi-year frost is a common problem for probing ecosystem-driven permafrost in the boreal region."

And to start the next paragraph:
"We differentiated three quasi-stable permafrost conditions and four degradation stages using a system modified from Jorgenson (2021) to address the complicated nature of permafrost formation and degradation in boreal ecosystem-driven permafrost (Supplemental Information Table 1; Supplemental Information Fig. 5)"

And this reference has been added:
Jorgenson, M. T. (2021). Thermokarst. In J. Schroeder (Ed.), Treatise on Geomorphology, 2nd Edition (Vol. Vol. 4 Cryospheric Geomorphology, pp. 1-22). Amsterdam, The Netherlands Elsevier. https://doi.org/10.1016/B978-0-12-818234-5.00058-4.

5.  Line 490: The limitations of each technology were discussed in detail; excellent work. However, the application of emerging ML is vague.
To address this we have added the following sentence at the end of 4. Discussion and Conclusions:
"Of particular interest are ways to combine these techniques with multiple airborne and spaceborne remote sensing products to identify relationships of key ground state variables that control permafrost stability like the depth of the snowpack, surface soil moisture, soil strength, surface water ponding, and seasonal subsidence."

6.  Conclusion: Among these technologies mentioned, what is the most promising one?
This is a good comment. To address it we have added this to the abstract:
"No single method provides all the information typically needed to adequately assess permafrost undergoing change. For example, frost probing yields insight into top-down thaw, LiDAR allows the identification of vertical and lateral subsidence, and ERT can identify the presence/absence of permafrost at 10s of meters depth."
We also added relevant text in Section 4.1 when we discussed the pros and cons of each method.

And in the Conclusions, 4.2 Degradation and Aggradation Stages we have inserted this paragraph (with an addition sentence added below based on a comment by this Reviewer.
"No single method provides all the information typically needed to adequately assess permafrost undergoing degradation or aggradation. For example, frost probing yields insight into top-down thaw or indicate areas where near-surface permafrost is aggrading upward but with limited spot measurements. LiDAR allows the identification of vertical and lateral subsidence upon thaw or heave associated with aggradation at a relatively larger spatial coverage but is limited to surface measurements. ERT can identify the presence/absence of permafrost at 10s of meters depth but is not as well suited for survey-level measurements of permafrost bodies whether they are stable or changing.

A combined use of three methods is more effective to characterize permafrost degradation at multiple dimensions than the application of each individual method. Though our study focused on sites in Interior Alaska the methods we applied here can be used to survey and track changes in other permafrost terrains."

7. The findings were based on the condition in Alaska. If such knowledge is being transferred to other cold regions, what is the most important issue for localization?
This is a good comment. To address this, we have added:
In 4. Discussion at the end of the first paragraph we have edited/added:
"Permafrost in Interior Alaska has been slowly thawing for the past ~500 years with sporadic periods of accelerated thaw typically attributed to wildfire and subsequent permafrost stabilization or aggradation associated with forest succession (Jorgenson et al. 2001; Jones et al., 2013). Air temperature increases since the 1970s in Interior Alaska (Osterkamp, 2005) and across the Arctic (Smith et al., 2022) have lead to increased permafrost temperatures and widespread thaw. Numerous recent studies in Interior Alaska show an acceleration of permafrost degradation with deeper seasonal thaw depths (Douglas et al., 2020; Euskirchen et al., 2024), widespread talik expansion (Farquharson et al., 2022), and an increased thermokarst development (Douglas et al., 2021; Minsley et al., 2022; Brodylo et al., 2024). Studies from sites across the Arctic show increasing soil temperatures (Chen et al., 2022) and subsidence due to permafrost degradation (Streletskiy et al., 2024)."

These references have been added:
Chen X, Jeong S, Park CE, Park H, Joo J, Chang D, Yun J. Different responses of surface freeze and thaw phenology changes to warming among Arctic permafrost types. Remote Sensing of Environment. 2022 Apr 1;272:112956.

Smith SL, O'Neill HB, Isaksen K, Noetzli J, Romanovsky VE. The changing thermal state of permafrost. Nature Reviews Earth & Environment. 2022 Jan;3(1):10-23.

Streletskiy DA, Maslakov A, Grosse G, Shiklomanov N, Farquharson LM, Zwieback S, Iwahana G, Bartsch A, Liu L, Strozzi T, Lee H. Thawing permafrost is subsiding in the Northern Hemisphere-review and perspectives. Environmental Research Letters. 2024 Dec 24.

In 4.1 Relative strengths and weaknesses of different permafrost degradation measurements:
"Though our study focused on sites in Interior Alaska the methods we applied here can be used to survey and track changes in other permafrost terrains."

In the last paragraph of the Conclusions, we have added:
"Given the positive and negative aspects of ground surface surveys, airborne LiDAR, and geophysical investigations a coupled application of these methods is warranted to track permafrost thaw at similar locations or other permafrost regions."

---

## Author Response (AR1)

Dear Editor and Reviewers-

Please find herein an addressal of the various comments and suggested edits for our manuscript. Oure **new text is in blue and bolded** and new/edited manuscript text is in red with quotations.

Dear authors and reviewers of "Comparing thaw probing, electrical resistivity tomography, and airborne lidar to quantify lateral and vertical thaw in rapidly degrading boreal permafrost".
The Editor's comments after the review are in green text.
I have gone through the comments and responses, and I would like to ask Reviewer Nr 2 to respond on the following three points, to see if they are satisfied. These points are as follows:

1) As a major point, Reviewer 2 asked for dates of the frost probing and even the field work. Knowing just the year is not sufficient.
I see that Reviewer 2 writes: "P4 L100, there is no mention of the exact timing or dates of the site visits that were completed each year from 2012 to 2020. This information is very important, as biases can be introduced if site visits occurred later in the thaw season as the study period progressed. Please provide a table of the dates of the site visits and additional contextual information, such as the sum of thawing degree days leading up to the site visit date for each year. This can be included as supplementary information if needed."

The authors' reply was : "As was stated earlier we have added the following text: "Repeated thaw probing using a metal rod in late summer (September) quantified changes in the top of near-surface permafrost. Maximum probing depths varied from 2.5 to 4.0 m depending on the number of extensions used, occurrence of gravel, and stickiness of unfrozen silts."
I agree that the dates are very important pieces of information. To the authors: Can a table of field work dates not be provided as a Supplemental table? Could the reviewer please comment on whether reference to simply "late September" is sufficient (note that above states only "September"), if the authors cannot provide the precise dates.

2) Could the reviewer please look at the new table about the class definitions and comment on whether this is acceptable? In addition, could the authors consider whether the information is complete and consistent here, eg, Shouldn't you include in S1 Table 1, the thaw depths that define your classes?

3) Is reviewer 2 satisfied with this reply from the authors?
P17 L352-354, many years of personal experience with frost probing in various materials would lead me to disagree with these statements. I would say that frost probing does not provide precise measurements, as the depth to the frost table may vary according to the person performing the probing, to the material, and to the timing of the measurement from year to year. I also do not think that it is possible to detect precise changes in soil texture with depth, and if this is the method that is used to determine stratigraphy, then soil pits need to be dug to verify these interpretations. The discussions of the limitations of probing on P18 L370-373 also contradict this description of probing as a "precise" method that can "detect marked changes in soil texture".

This is a great point and we have addressed it by clarifying WHERE the frost probe measurements are most useful and HOW they have been confirmed with soil pits and cores: "Thaw probing, accompanied by surveying of ground and water-surface elevations, yielded the most precise measurements of subsurface changes in the permafrost table than ERT or repeat LiDAR and this helped identify degradation stages (Figs. 2 and 3). Probing can detect changes in soil texture (peat, silt, sand, gravel) with depth, however, soil stratigraphy was confirmed with soil pits and boreholes (Brown et al., 2015)."
If the Reviewers are unsatisfied with any other changes, please let the editor know.
* * *
Now there are some Editor's comments (which are in green color) regarding the authors' replies.
*1) In your changes to the manuscript based on Reviewer comments, there may be some language to correct (see my suggestion below in red). In fact, in a few of the additions or changes to text in the Reply to Reviewers. Please do some grammatical checking before submitting the next version.*

"A total of 183 Ground Control Points (GCPs) were measured using real time kinematic and post processed kinematic techniques to validate the accuracy of the LiDAR DEM products. The average error was 0.093 m in the DEM (95th percentile) across different ecotypes (Zhang et al., 2023), leading to the propagated error of 0.13 m for the differenced LiDAR DEM. This error was considered in our thaw stage analysis."
**We have made the suggested edits.**

2) In your reply to reviewer 2, I did not see an answer to this. Is this already addressed?
Given that the first ERT survey and the first LiDAR acquisition were conducted in different years, it would be very helpful if the authors could provide a description of the climatic context for the 2012 versus 2014 years. The authors should also include a limitations section in the Discussion that addresses overall limitations to their study, including issues in comparing between methods in different years.
**We apologize for not addressing this directly. The other Reviewer made the following comment:**

"P4 L76-81, please consider providing additional context related to climatic conditions during the study period. A plot showing mean annual air temperatures and total annual precipitation from 2012 to 2020 would help to contextualize the study.
We cite Jorgenson et al. (2020) repeatedly and this paper provides wonderful summaries of temperature and precipitation in Figure 12 that includes most of the time suggested."

**And we addressed it by adding the following to the manuscript in the area where this Reviewer called for it:**
"More detailed information on the regional climate, with a focus on Tanana Flats, is provided in Supplemental Information Figs. 1 and 2. The period between 2012 and 2020 (when the majority of the field measurements were collected, including repeat electrical resistivity tomography) included three of the ten warmest/wettest summers and three of the ten warmest wettest winters in the ~100-year record. However, no top ten summers or winters for cool and dry or cold and low snow, favorable to permafrost stability or aggradation, occurred during the same timeframe. Two of the five wettest summers in the entire meteorological record (2014 and 2016) and two of the top three highest mean annual air temperatures (MAATs) were also recorded during the study period. This provided conditions favorable to permafrost thaw."

3) The authors have added "Based on decadal mean annual temperatures at the Fairbanks International Airport the area warmed ~2.3 °C between the 1930s-1940s and 2010-2020. Over that same timeframe mean summer temperatures (May 1 to October 10) warmed ~1.7 °C while mean winter temperatures (October 11-April 30) warmed 2-4 °C (Douglas et al., 2024). "

This is an addition of text, so I will comment on this. All of the temperatures are given as a mean value change, but for winter temperatures a range is given. Why? Please clarify what the range refers to.
**We propose to change this to:**
"Based on decadal mean annual temperatures at the Fairbanks International Airport the area warmed ~2.3 °C between the 1930s-1940s and 2010-2020. Over that same timeframe mean summer temperatures (May 1 to October 10) warmed ~1.7 °C while mean winter temperatures (October 11-April 30) warmed ~3 °C (Douglas et al., 2024)."

4) Grammatical issue here: "We estimate the accuracy to be mostly with 10 cm in open areas and 30 cm in forested areas."
**We have changed this to:**
"We estimate the accuracy to be mostly within 10 cm in open areas and 30 cm in forested areas."

5) Please give the full reference, if this is a book: Blaschke, T., S. Lang, E. Lorup, J. Strobl, and P. Zeil. 2000. Object-Oriented Image Processing in an Integrated GIS/Remote Sensing Environment and Perspectives for Environmental Applications. Environmental Information for Planning, Politics and the Public 2: 555–570.
**Would this provide the required information:**
Blaschke, T., S. Lang, E. Lorup, J. Strobl, and P. Zeil. 2000. Object-Oriented Image Processing in an Integrated GIS/Remote Sensing Environment and Perspectives for Environmental Applications. In: Environmental Information for Planning, Politics and the Public, 2(1995), October 2000: 555–570.

6) In response to the reviewer comments on the repetition on P7, L173-187, the authors add

"We differentiated trends across each transect into seven degradation/aggradation stages (Supplemental Information Table 1). "
I think you have six stages, and not seven now?
**Yes. Good catch and thank you. This has been changed to:**
"We differentiated trends across each transect into six degradation/aggradation stages (Supplemental Information Table 1)."

7) I agree with reviewer 2 here. You cannot know what happened between 2012 and 2014 if you do not back it up with data. Either point to the data that supports the statement or remove it.
Review 2 writes: P15 L299-300, this statement that the resistivity values decreased between 2012 and 2014 should be removed if the authors do not present the tomograms from 2014 in this paper.
**We sincerely apologize for the confusion here. The "2014" was supposed to be "2020" and we have changed it here. These are indeed the only two years for which we have the resistivity measurements.**
"Resistivity values decreased between 2012 and 2020 and these correspond with the thaw degradation stages identified earlier."

8) Some grammar to fix here. P17 L333, "These are the most common landforms above boreal discontinuous permafrost" – what does this mean? That the permafrost plateaus mentioned in the previous sentence are the most common landforms? Please clarify.

My suggestion (feel free to modify) for a grammatical change is: "Plateaus are the most common landforms occurring in boreal discontinuous permafrost, and are a signature indicator of the presence of permafrost in the region."
**This has been changed as suggested.**

9) This addition could be written better for flow: "This is likely due to the lower ice content of lenticular ice from 1.2 to 2.8 m depth at TF88 compared to braided (reticulate-platy) ice from 1.5 to 2 m at T1 (Brown et al., 2015)."
**This has been changed as suggested.**

10) I cannot find a real answer to the reviewer's question in the following:
Reviewer 2 writes: CONCLUSION P22 L494, what data was provided from 2004? The methods section describes the survey transects as being established in 2011/2012, not 2004? Please revise and correct.
You need to revise the sentence in the conclusion, and it seems you replace this:

"Between 2004 and 2020 top-down thaw of near surface permafrost doubled from 18% to 36%."
With this:
"For example, at site T1, our control transect, top-down thaw of near surface permafrost occurred over 44% of our study sites between 2004 and 2020."
Which does not change the problem as I see it. What the authors say they have added to the methods for clarity (below) has not added any information about 2004, as the reviewer asked.
We have updated the Methods to clarify when each transect was initially surveyed or modified (expanded): "The Tanana Flats lowland has experienced numerous wildfires and we established transects to represent high severity fires in the summers of 1988 (TF88, 64.734 °N, 147.826 °W), 2001 (TF01, 64.644 °N, 148.295 °W), and 2010 (TF10, 64.716 °N, 148.010 °W). Transects were initially studied to assess the effects of fire on permafrost (Nossov et al., 2013): TF88 (200 m; first surveyed in 2012) is in an area burned in ~1950 that reburned in 1988; TF01 (established as a 320 m transect in 2012) is in an area burned in 2001, and TF10 (initially 100 m in 2011, extended to 200 m in 2012) is in an area burned in 2010. Transects were positioned to cross a range of permafrost and non-permafrost ecotypes and TF01 and TF10 were extended to more adequately represent different cover types. We also included an unburned site (T1, 64.722 °N, 147.959 °W) that has not burned in recent years (~1950s-present) for comparison. Supplemental Information Figs. 3 and 4 provide photographs of the field sites. T1 (initially 100 m in 1995, extended to 200 m in 2012) was established during ecological land surveys (Jorgenson et al. 1999). In Douglas et al. (2016) T1 is referred to as "1930". Two of the sites (T1 and TF88) were relatively ice-rich with thick peat and silts extending down 34 m whereas the other two study sites (TF01 and TF10) had sand and gravel at relatively shallow depths and lower ice contents."

We have updated Figures 2 and 3 to provide the date of any ground surface measurements.
To amplify this, we have added a mention of this where Figures 2 and 3 are introduced:
**"3 Results**
**3.1 Degradation stages identified along the transects**
We compiled the repeat surveys of surface topography, water depths, thaw depths, and ancillary subsurface measurements from the four transects to define trends of subsurface thaw and corresponding ground surface settlement between 2012 and 2020 (Figs. 2 and 3). Dates when ground surface elevations and thaw depth were measured are included in these Figures."

11) Editor comment:
Line 146: Airborne lidar surveys were conducted by Quantum Spatial Incorporated (Anchorage, Alaska) in May 2014 and 2020 in a period of 95% snowmelt with leaf-off conditions using a Lecia ALS laser system, **leading to an average pulse density larger than 25 points/m2** over the targeted area (~40 km2, Figure 1) where our field transects were located.
A lot of numbers are larger than 25 ... could you be more specific?

In section **2.2.2 Airborne LiDAR lidar data acquisition and analysis**
We have edited the text to say:
"Both datasets were collected at a high point density (larger than 25 points/m2) in a period of 95% snowmelt with leaf-off conditions. From this we generated very fine resolution lidar LiDAR DEM products were generated at 0.25 m resolution."
Does this clear up the question?

12) Editor comment: Do you need to change Figure 4 to show six classes rather than 8 no
**A new Figure 4 has been developed with n7 total classes( the 6 degradation ones and "undegraded.")**

---

## Author Response (AR2)

Dear Editor and Reviewers-

Please find herein an addressal of the various comments and suggested edits for our manuscript. Oure **new text is in blue and bolded** and new/edited manuscript text is in red with quotations.

**Public justification (visible to the public if the article is accepted and published)**:
Since several paragraphs were added to the latest version that have some grammar to be corrected or clarified, please make the following minor changes.

- Throughout document, "earth's" should be "Earth's"
Two changes were made:
First sentence of the abstract:
"Permafrost thaw across Earth's high latitudes is leading to dramatic changes in vegetation and hydrology."

First sentence of the Introduction:
"Permafrost is warming and degrading across Earth's high latitudes"

- Ln 72 "...quantify and monitoring..." should be "...quantify and monitor..."
This has been changed, as suggested, to:
"In this study we focus on how best to quantify and monitor permafrost degradation rather than assess the drivers of change and ecological changes."

- Ln 92 - Does the journal use the spelling "aeolian" or "eolian"?
Based on this paper:
Amory C, Trouvilliez A, Gallée H, Favier V, Naaim-Bouvet F, Genthon C, Agosta C, Piard L, Bellot H. Comparison between observed and simulated aeolian snow mass fluxes in Adélie Land, East Antarctica. The Cryosphere. 2015 Jul 30;9(4):1373-83
WQhich has:
"Aeolian snow transport events were correctly reproduced with the right timing and a good temporal resolution at both locations except when the maximum particle height was less than 1 m."

The journal uses: "aeolian"
We have made this edit to:
"Much of the area has a typical stratigraphic sequence of peat (0.5-1.5 m), aeolian silt (2-3 m), and alluvial sand and gravel (Jorgenson et al., 1999; Brown et al. 2015)."

- Ln 94 - "Excess ice contents in birch forests can be greater than 50% while ice contents in the black spruce stands are typically closer to 20% (Brown et al., 2015; Jorgenson et al., 2001; Jorgenson et al. 2025; Walters et al., 1998). " - change "contents" to "content", and consider modifying the sentence to reflect that it is the permafrost ground you are referring to and not the forests (trees).
We have changed this to:

"Permafrost is mainly epigenetic and formed during downward freezing. Excess ice contents in permafrost below birch forests can be greater than 50% while ice contents in the black spruce stands are typically closer to 20% (Brown et al., 2015; Jorgenson et al., 2001; Jorgenson et al. 2025; Walters et al., 1998)."

- Ln 97, 115, etc - Note where you can use the abbreviation ERT as you have already defined it.
We have changed it in two locations:
"Deep boreholes found permafrost at depths ranging from 7.3 m (Ferrick et al., 2008) to 47 m (Jorgenson et al., 2001), while ERT indicated minimum thicknesses of >20 m were common (Douglas et al., 2015)."

"The period between 2012 and 2020 (when the majority of the field measurements were collected, including repeat ERT) included three of the ten warmest/wettest summers and three of the ten warmest wettest winters in the ~100-year record."

We have not changed "electrical resistivity tomography" to "ERT" where it is present in section titles, however, please let us know if you wish to make this change. There are two of these instances.

- Ln 158 - "...to be within 3 cm..."
This has been changed, as suggested, to:
"As the soils within the top 3 m were mostly peat and silt, shallow (<1 m) probing typically hit a hard refusal boundary (indicating frozen conditions) for stable permafrost; for these conditions we consider the accuracy to be within 3 cm (given a "soft" mossy and litter-rich ground surface)."

- Line 160 "...to be within 20 cm..."
This has been changed, as suggested, to:
"Occasionally, in partially degrading permafrost near the surface (within 1.5 m) the probe encountered frozen ground with substantial unfrozen water content with refusal gradually becoming harder across a ~20 cm transition zone; we consider the accuracy under these conditions to be within 20 cm."

- Line 165 - There is quite a bit of new text here, and I am a bit confused as the paragraph leads in with reference to probing, but here switches to thickness of the permafrost/seasonal frost layer. Please have a careful look at this paragraph and revise for clarity.
To reduce uncertainty, we have changed the information on seasonal frost to a separate paragraph and clarified it a little bit:
"Differentiating persistent seasonal frost from very thin permafrost is a challenge, particularly when only frost probe measurements are available (i.e. no boreholes). Here we are confident that frost <30 cm thick at the end of summer is seasonal frost and frost >50 cm thick is permafrost; for in between thicknesses we have low confidence as to whether it is seasonal or permanent frost. In some areas we noted a thin frost layer (30 to 50 cm) to persist for 2 to 3 seasons, and we refer to this as multi-year frost. This multi-year frost is a common problem for

probing ecosystem-driven permafrost in the boreal region."

- Line 220 - " The average error was 0.093 m..." indicate in this sentence whether you are referring to vertical or other error.
This has been changed to:
"The average vertical elevation error was 0.093 m in the DEM (95[th] percentile) across different ecotypes (Zhang et al., 2023), leading to the propagated error of 0.13 m for the differenced LiDAR DEM."

- Line 247 - Please check the grammar of this line. "DI thaw depths >100 cm and <120 cm or had increased by >30 cm as a brief transitional stage ..." I think it is poorly phrase, or maybe a word missing after "or".
This was hard to follow for sure. We have changed it to reflect the other location where "degradation-initial" is described:
"DI (thaw depths ~ 1.1 m or had increased by >30 cm as a brief transitional stage) occurred along 36% of the transects."

Line 250 - remove the , after the ) - In fact, please double check the grammar of this paragraph. I find it difficult to read.
We have edited/clarified a few areas in the paragraph:
"We compiled the repeat surveys of surface topography, water depth, thaw depth, and ancillary subsurface measurements from the four transects to define trends of subsurface thaw and corresponding ground surface settlement between 2012 and 2020 (Figs. 2 and 3). Dates when measurements were collected are included in these Figures. We differentiated trends across each transect into six degradation/aggradation stages (Supplemental Information Table 1). UD (thaw depths remained <100 cm with annual changes <30 cm) occurred along 29% of the transects in 2012 and decreased to 8% by 2020 (Figs. 2 and 3). DI (thaw depths ~ 1.1 m or had increased by >30 cm as a brief transitional stage) occurred along 36% of the transects. DPS (vertically increasing thaw depths >120 cm but not more than ~250-300 cm), increased from 12% to 20% between 2012 and 2020. DPD (thaw depths increased to >250-300 cm) increased from 6 to 16%. Together, these shallow and deep progressive degradation stages indicating top-down thaw of near-surface permafrost increased from 18% to 36% over the 8 year study period. We presume DPD denotes areas with open taliks. DL (lateral thaw bogs and fen margins) increased slightly from 3% to 6%. Regions characterized as DCO occurred under old bogs and fens and presumably had completed degradation through the entire permafrost zone to form through taliks. They increased slightly between 2012 and 2020 from 35% to 44%. We attribute this change, however, due to loss of Repeat-Permafrost-Thin (RPT) and MF (Figs. 2 and 3) near the surface. MF was used to differentiate areas with a thin layer (typically <30 cm) of frozen ground that persisted for 14 years (permafrost is conventionally defined as frost persisting more than 2 years). MF decreased from 7% to 0%."

Line 374 - " For example, frost probing yields insight into top-down thaw or indicate areas..." should be "indicates".
This has been changed, as suggested, to:

"For example, frost probing yields insight into top-down thaw or indicates areas where near-surface permafrost is aggrading upward but with limited spot measurements."

Line 377 - "...permafrost at 10s of meters ..." should be "... permafrost at tens of meters..."
This has been changed, as suggested, to:
"ERT can identify the presence/absence of permafrost at 10s tens of meters depth but is not as well suited for survey-level measurements of permafrost bodies whether they are stable or changing."

Line 378 - The authors added the statement that ERT "...is not as well suited for survey-level measurements of permafrost bodies whether they are stable or changing." Please look carefully at this statement and see if you need to modify it, such as "... .is not as well suited for survey-level measurements of vertical or lateral degradation of permafrost bodies." (assuming this is what you mean).
Thank you for this suggestion. We have changed it but we added "as frost probing":
"ERT can identify the presence/absence of permafrost at tens of meters depth but is not as well suited as frost probing for survey-level measurements of vertical or lateral degradation of permafrost bodies."

Also, Make sure all of the Supplemental information mentioned in your MS are present in the Supplemental information. I see only one Table there now.
There are five Figures and two Tables in the SI and they have been uploaded with this most recent version.